# Reproducibility Study of 'SLICE: Stabilized LIME for Consistent Explanations for Image Classification'

**Aritra Bandyopadhyay**                    *aritra.bandyopadhyay@student.uva.nl*
*University of Amsterdam*
*Informatics Institute*

**Chiranjeev Bindra**                    *chiranjeev.bindra@student.uva.nl*
*University of Amsterdam*
*Informatics Institute*

**Roan van Blanken**                    *roan.van.blanken@student.uva.nl*
*University of Amsterdam*
*Informatics Institute*

**Arijit Ghosh**                    *arijit.ghosh@student.uva.nl*
*University of Amsterdam*
*Informatics Institute*

**Reviewed on OpenReview:** *https://openreview.net/forum?id=vKUPXuEzj8&noteId=IgCeCiP7N6*

## Abstract

This paper presents a reproducibility study of SLICE: Stabilized LIME for Consistent Explanations for Image Classification by Bora et al. (2024). SLICE enhances LIME by incorporating Sign Entropy-based Feature Elimination (SEFE) to remove unstable superpixels and an adaptive perturbation strategy using Gaussian blur to improve consistency in feature importance rankings. The original work claims that SLICE significantly improves explanation stability and fidelity. Our study systematically verifies these claims through extensive experimentation using the Oxford-IIIT Pets, PASCAL VOC, and MS COCO datasets. Our results confirm that SLICE achieves higher consistency than LIME, supporting its ability to reduce instability. However, our fidelity analysis challenges the claim of superior performance, as LIME often achieves higher Ground Truth Overlap (GTO) scores, indicating stronger alignment with object segmentations. To further investigate fidelity, we introduce an alternative AOPC evaluation to ensure a fair comparison across methods. Additionally, we propose GRID-LIME, a structured grid-based alternative to LIME, which improves stability while maintaining computational efficiency. Our findings highlight trade-offs in post-hoc explainability methods and emphasize the need for fairer fidelity evaluations. Our implementation is publicly available at our GitHub repository.

## 1 Introduction

Explainability in AI is crucial as deep learning models grow increasingly complex Grobrügge et al. (2024), yet interpreting their non-transparent decisions remains challenging. Post-hoc methods like Local Interpretable Model-agnostic Explanations (LIME)(Ribeiro et al., 2016) address this by approximating a model's local behavior with simpler surrogates. LIME generates perturbed samples around an input and fits a linear model, but while locally faithful, its explanations often suffer from inconsistency across runs(Gosiewska & Biecek, 2019; Zafar & Khan, 2019; Zhao et al., 2021; Zhou et al., 2021).

Stabilized LIME for Consistent Explanations (SLICE) (Bora et al., 2024) aims to resolve LIME's instability using adaptive Gaussian blurring and sign entropy-based feature elimination (SEFE). This reproducibility

study validates SLICE's claims regarding improved consistency and fidelity by replicating its core experiments using the Oxford-IIIT Pets, PASCAL VOC, and MS COCO datasets. Our implementation is available at [1].

Beyond verifying SLICE's claims, this study investigates limitations and proposes alternatives. Observing LIME's instability stems partly from segmentation randomness and SLICE's significant computational cost, we introduce GRID-LIME. It employs a structured grid segmentation, optimized using model predictions, to enhance stability while maintaining efficiency closer to LIME. Furthermore, recognizing that perturbation-based fidelity metrics like AOPC can be biased by the perturbation strategy (e.g., SLICE's blur vs. LIME's masking), we introduce the Ground Truth Overlap (GTO) metric. GTO directly compares explanations to ground-truth segmentations, offering a potentially less biased fidelity assessment. Our findings confirm SLICE's improved consistency but challenge its superior fidelity claims, particularly when considering GTO scores and computational cost, highlighting inherent trade-offs in post-hoc explainability. This work details the reproduction of SLICE's claims, introduces and evaluates GRID-LIME and GTO, analyzes SLICE's components, and extends evaluations across multiple datasets.

## 2    Scope of reproducibility

To address the stability issues in LIME-based explanations, Bora et al. (2024) proposed SLICE, a method that enhances LIME by incorporating a novel feature elimination strategy and an adaptive perturbation technique. Specifically, SLICE utilizes Sign Entropy-based Feature Elimination (SEFE) to remove superpixels with high sign entropy, ensuring greater consistency in explanations. Additionally, SLICE introduces an adaptive Gaussian Blur mechanism that dynamically selects the optimal perturbation level, reducing variance in feature importance rankings and improving the overall reliability of explanations.

The focus of our work is on reproducing the following claims made by Bora et al. (2024):

- **Claim 1:** LIME suffers from inconsistencies in its explanations, primarily due to variance in sign and importance ranking of superpixels. These inconsistencies stem from the perturbation method used by LIME.

- **Claim 2:** The Sign Entropy-based Feature Elimination (SEFE) method improves stability in explanations by identifying and eliminating spurious superpixels.

- **Claim 3:** The introduction of Gaussian Blur with an adaptively selected hyperparameter $\sigma$ enhances the perturbation method, leading to greater consistency in explanations.

- **Claim 4:** Explanations generated by SLICE demonstrate significantly better fidelity compared to those produced by LIME.

## 3    Post-Hoc Explainability Methods

Post-hoc explainability methods aim to interpret complex AI models without requiring access to their internal structures. Local Interpretable Model-agnostic Explanations (LIME) (Ribeiro et al., 2016) is a widely used method that approximates a model's decision function locally by generating perturbed samples around a given instance and fitting a linear surrogate model. However, LIME suffers from inconsistencies across different runs, leading to instability in feature importance rankings and sign variations.

Several extensions have been proposed to improve the stability of LIME. **DLIME** (Rashid et al., 2024) employs hierarchical clustering to select perturbed samples from the cluster nearest to the instance of interest, ensuring greater locality adherence. **S-LIME** (Shi et al., 2020) leverages a hypothesis testing framework based on the Central Limit Theorem to determine the number of samples required for stable explanations. **ALIME** (Knab et al., 2025) introduces an autoencoder-based weighting function to improve coefficient stability by refining sample selection. **BayLIME** (Dehghani et al., 2024) formulates explanations as a Bayesian-weighted sum of prior knowledge and new observations, reducing variance in feature importance.

---

[1] https://github.com/CSBXAI/slice_reproducibility

More recent methods focus on enhancing LIME's applicability to image-based models. **Beyond Pixels** (Knab et al., 2025) integrates hierarchical feature representations with segmentation foundation models to improve visual explanations. **SMILE** (Dehghani et al., 2024) employs statistical distances based on empirical cumulative distribution functions to enhance interpretability in instruction-based image editing tasks. These advancements contribute to improving LIME's robustness across diverse applications.

As our study primarily investigates post-hoc explanations for image classification, we focus on evaluating **SLICE** (Bora et al., 2024), which introduces an adaptive Gaussian perturbation method and a feature elimination strategy to improve stability. For SLICE and LIME, we provide a comprehensive assessment of its effectiveness in producing consistent and reliable explanations.

## 4 Methodology

The objective of this reproducibility study is to verify the stability and consistency of SLICE, which uses Adaptive Gaussian Blur and Sign Entropy based Feature Elimination (SEFE). For re-implementing the code in PyTorch, we referred to the SLICE code that was made publicly available by the authors at [2].

It should be noted that the author's code is not referenced directly in the original paper. However, since the link to the author's code was provided to us by the authors, we decided to use this implementation as our reference. Furthermore, we added intermediate images at every step of the process, making the interpretation pipeline more transparent. We also introduced a new interpretability metric to enhance the usability of SLICE. The code for this reproducibility study is publicly available on Github [3]

### 4.1 Model descriptions

#### 4.1.1 LIME

Local Interpretable Model-agnostic Explanations (LIME) is a widely used post-hoc interpretability method that approximates the decision boundary of black-box models using a local surrogate model. LIME operates through three key steps: perturbation, weighting, and surrogate model training. Given a black-box model $f$ and an input instance $x$, LIME generates perturbed images $z$ by masking superpixels. A weighting function $\pi_x$ assigns importance to these perturbed samples based on their similarity to the original input. Subsequently, a linear surrogate model $g$ is trained on these perturbed samples $z'$ to approximate $f$ locally. $\mathcal{Z}$ is the set of all perturbed samples. The objective function for LIME is given by:

$$\mathcal{L}(f, g, \pi_x) = \sum_{z,z' \in \mathcal{Z}} \pi_x(z) \left( f(z) - g(z') \right)^2 \tag{1}$$

While LIME effectively provides locally faithful explanations, it suffers from instability in feature importance rankings and sign fluctuations across multiple runs, limiting its reliability.

#### 4.1.2 SLICE

SLICE (Stabilized LIME for Consistent Explanations) is designed to mitigate the instability of LIME by introducing two key components: Adaptive Gaussian Blur and Sign Entropy-based Feature Elimination (SEFE). An overview of SLICE is illustrated in Figure 1.

The first component, Adaptive Gaussian Blur, applies a range of perturbation strengths ($\sigma$ values) to superpixels using Gaussian blurring. Instead of replacing superpixels with a fixed value (as in LIME), SLICE adaptively selects an optimal $\sigma$ value based on entropy, ensuring perturbations are closer to the original data distribution.

---

[2] https://github.com/rebathip/SLICE-Stabilized-LIME-for-Consistent-Explanations-for-Image-Classification/tree/main

[3] https://github.com/CSBXAI/slice_reproducibility

The second component, SEFE, eliminates superpixels that exhibit high sign entropy, meaning their attribution flips between positive and negative across multiple iterations. This ensures greater stability in feature importance rankings and improves the consistency of explanations.

The SLICE algorithm iteratively generates $N$ perturbed samples and trains $M$ Ridge Regression models. SEFE identifies and removes unstable superpixels, while the final ranking of superpixels is based on the learned coefficients from the regression models. Training continues until a predefined number of iterations or a stability tolerance threshold is reached.

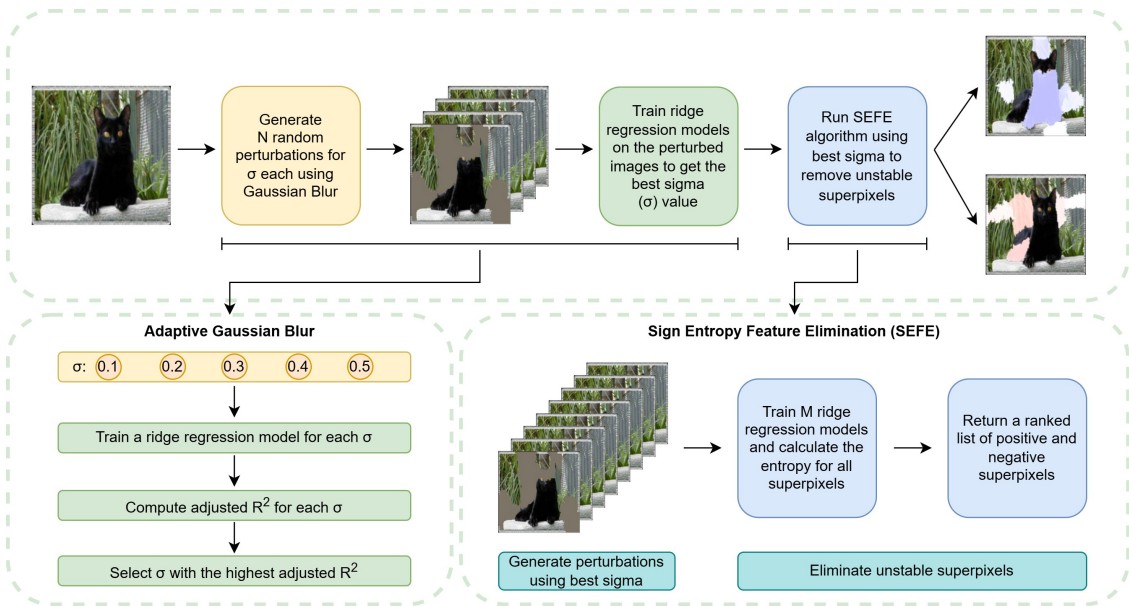

Figure 1: Overview of the SLICE framework. SLICE improves LIME's stability by incorporating Adaptive Gaussian Blur and Sign Entropy-based Feature Elimination (SEFE).

### 4.1.3 GRID-LIME

`Quickshift` generates superpixel segments based on the intensity values of the three color channels of the image, introducing variability across runs and potentially ignoring semantically relevant boundaries captured by the model. To address this and LIME's segmentation instability, we introduce GRID-LIME. This method replaces `Quickshift` with a structured grid-based segmentation where the optimal grid size is determined using the black-box model's predictions, aiming to improve consistency without SLICE's computational overhead.

As shown in Figure 2, GRID-LIME evaluates a set of potential grid sizes, $S = \{s_1, s_2, ..., s_k\}$. For each grid size $s \in S$, the image $x$ is divided into non-overlapping grid cells $c_{s,j}$. Perturbed images $x_{s,j}$ are generated by setting the pixels within each cell $c_{s,j}$ to zero (or another baseline value). Let $f(x)$ be the model's prediction probability for the target class on the original image and $f(x_{s,j})$ be the prediction on the perturbed image where cell $c_{s,j}$ is masked.

To select the optimal grid size, we analyze the sensitivity of the model's predictions to these grid-based perturbations. For each grid size $s$, we calculate the prediction differences $\Delta_{s,j} = |f(x) - f(x_{s,j})|$ for all cells $j$. We then compute the mean $\mu_s$ and standard deviation $\sigma_s$ of these differences $\{\Delta_{s,j}\}_j$. The Coefficient of Variation (CV) for grid size $s$ is calculated as:

$$\text{CV}_s = \frac{\sigma_s}{\mu_s} \tag{2}$$

The optimal grid size $s^*$ is chosen as the one maximizing the CV, indicating the grid resolution that yields the highest relative variability in prediction changes upon perturbation:

$$s^* = \underset{s \in S}{\operatorname{argmax}} \; \mathrm{CV}_s \tag{3}$$

This process ensures a balance between segmentation granularity and stability, selecting a grid that is informative according to the model's sensitivity. Once $s^*$ is determined, GRID-LIME proceeds with the standard LIME framework (perturbation, weighting, surrogate model training) using the superpixels defined by the optimal grid $s^*$. This structured, model-informed segmentation helps reduce the randomness associated with LIME's original segmentation step. However, a limitation remains: the rigid grid structure may not always align perfectly with natural object boundaries, potentially impacting interpretability for irregularly shaped objects compared to adaptive segmentation methods. The specific range of grid sizes tested in our implementation is detailed in Appendix C.

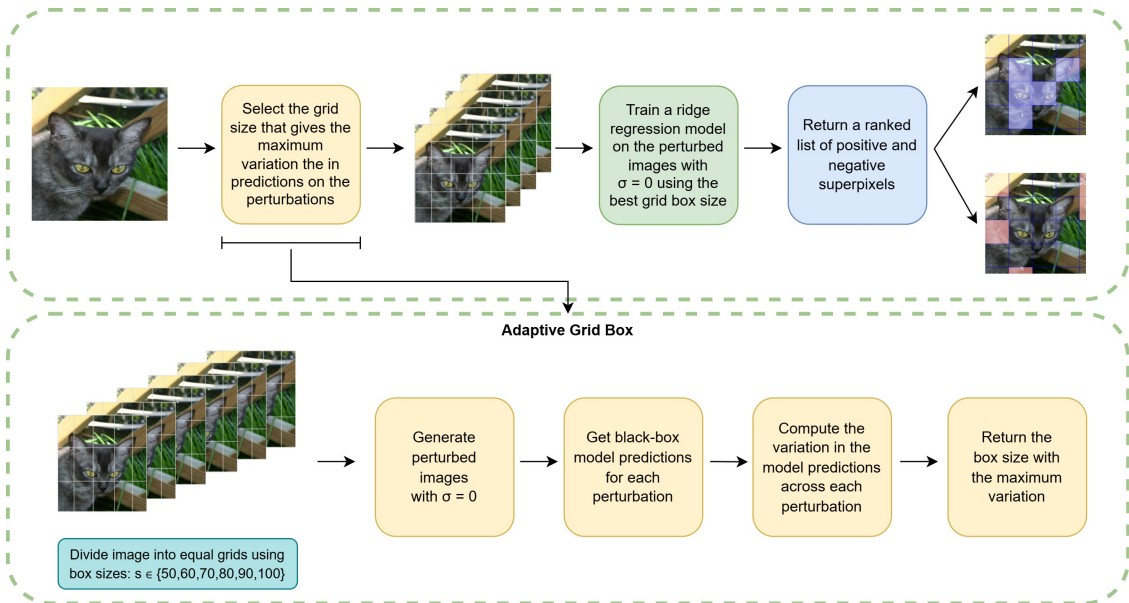

Figure 2: Illustration of GRID-LIME. GRID-LIME replaces color-space-based segmentation with a grid-based approach to ensure stable explanations. Here, $\sigma = 0$ denotes that it uses the same perturbation strategy as LIME.

## 4.2 Datasets

To evaluate the reproducibility of SLICE and compare its performance against LIME, we conduct experiments using three widely recognized image classification datasets: **Oxford-IIIT Pet Dataset**, **PASCAL Visual Object Classes (VOC)**, and **MS COCO**. These datasets provide a diverse range of images, enabling a comprehensive assessment of explanation consistency and fidelity.

### 4.2.1 Oxford-IIIT Pet Dataset

The Oxford-IIIT Pet dataset (Parkhi et al., 2012) consists of 37 categories of pet images, covering various breeds of cats and dogs. Each category contains approximately 200 images, leading to a total of 7,349 images. The dataset includes pixel-wise trimap segmentation annotations, enabling precise localization of object regions within the images. This dataset is particularly useful for evaluating explanation methods due to its clear object boundaries and distinct features. In our study, we randomly sample 50 images from this dataset and apply LIME, SLICE, and GRID-LIME to analyze the consistency and fidelity of their explanations.

### 4.2.2  PASCAL VOC 2009

The PASCAL Visual Object Classes (VOC) 2009 dataset (Everingham et al., 2010) is a benchmark dataset for object recognition tasks. It contains a total of 7054 images in the training and validation set. The dataset includes 20 object categories, covering a broad range of natural and man-made objects, making it suitable for testing explainability methods across different object types. Each image is annotated with bounding boxes and pixel-wise segmentation masks. Given its complexity and variation in object scales, we evaluate whether SLICE and GRID-LIME can improve the stability of LIME explanations. For our experiments, we randomly select 50 images from this dataset for comparative analysis.

### 4.2.3  MS COCO

The Microsoft Common Objects in Context (MS COCO) dataset (Lin et al., 2015) is one of the largest and most diverse datasets for image recognition and segmentation tasks. It contains over 330,000 images, annotated with 80 object categories, instance-level segmentation masks, and dense captions. This dataset presents a greater challenge due to the presence of multiple objects per image, occlusions, and background clutter, making it an ideal testbed for evaluating robustness in explanation methods. We apply LIME, SLICE, and GRID-LIME to 50 randomly selected images from the validation set to assess their performance in complex visual environments.

### 4.3  Hyperparameters

To ensure a fair and reproducible comparison between SLICE, LIME, and GRID-LIME, we carefully tune hyperparameters based on prior research and controlled experiments. Each explainer algorithm (SLICE, LIME, or GRID-LIME) is executed for 10 iterations per image to generate the results presented in Section 5.

For SLICE, we used the authors' choice of hyperparameters for all of the experiments. We train 1000 Ridge regression models to obtain model coefficients for all superpixels. Each Ridge model is trained on 500 randomly generated perturbations of the original test image. The Adaptive Gaussian Blur mechanism in SLICE requires selecting the optimal $\sigma$ value. We conduct a small-scale hyperparameter search, testing values in the range $[0.1, 0.5]$ with a step size of 0.1, and select the $\sigma$ value yielding the highest adjusted $R^2$.

In the Sign Entropy-based Feature Elimination (SEFE) module, the tolerance limit is set to 3, and the maximum number of iterations is capped at 10 to balance computational efficiency and convergence.

GRID-LIME, our proposed variant, does not require a segmentation algorithm such as quickshift. Instead, it utilizes structured grid-based segmentation, where we empirically determine the optimal grid size by evaluating perturbation consistency across different configurations.

For all experiments, we randomly sample 50 images per dataset and apply the same hyperparameters across SLICE, LIME, and GRID-LIME to maintain consistency. The details of hyperparameter values used in our experiments are provided in Appendix C.

### 4.4  Experimental setup and code

In this study, we conduct experiments to evaluate the performance of SLICE, LIME, and GRID-LIME in generating interpretable explanations. Our methodology and hyperparameter values follow the original paper, ensuring reproducibility. Additionally, we introduce an auxiliary experiment to assess the alignment of saliency maps with ground-truth segmentation maps. For MS COCO and PASCAL VOC datasets, we sample those images that have only one segmentation object map present for evaluation using Ground Truth Overlap (GTO).

We use ResNet50 (He et al., 2016) and InceptionV3 (Szegedy et al., 2016) models, both pretrained on ImageNet as used in the original paper. For superpixel segmentation, we apply `quickshift` (Vedaldi & Soatto, 2008) from the `skimage` library. SLICE iteratively perturbs images by blurring selected superpixels, fits local ridge regression models, and eliminates inconsistent superpixels until convergence. The Gaussian blur parameter $\sigma$ is optimized via a small hyperparameter search.

To evaluate model explanations, we use multiple metrics assessing consistency and fidelity. Consistency is measured through Average Sign Flip Entropy (ASFE), which quantifies sign stability across multiple runs, and Average Rank Similarity (ARS), which evaluates the consistency of ranked importance scores. We combine these into the Combined Consistency Metric (CCM), a metric introduced by the original paper, providing a unified measure of explanation stability.

Fidelity is assessed using the Area Over Perturbation Curve (AOPC), which tracks classifier confidence changes as important superpixels are perturbed, and the Area Under Curve (AUC), which captures the cumulative impact of these perturbations. Higher AOPC and higher AUC values for insertion and higher AOPC and lower AUC values for deletion of superpixels indicate that the explainer correctly identifies the most influential features.

As an additional metric, we introduce Ground Truth Overlap (GTO) to evaluate the alignment between model-generated saliency maps and ground-truth segmentation maps. Unlike previous metrics relying on classifier confidence, GTO directly compares the predicted importance regions with known object boundaries. Figure 3 illustrates the computation of the GTO metric and provides an example of its calculation.

$$GTO = \frac{\text{Area of Ground Truth} \cap \text{Area of top-k positive superpixels}}{\text{Area of top-k positive superpixels}} \tag{4}$$

Lastly, our code is available on GitHub[4]. A detailed analysis of the computational requirements and environmental impact of our experiments, including runtime and estimated carbon emissions, is provided in Appendix D.

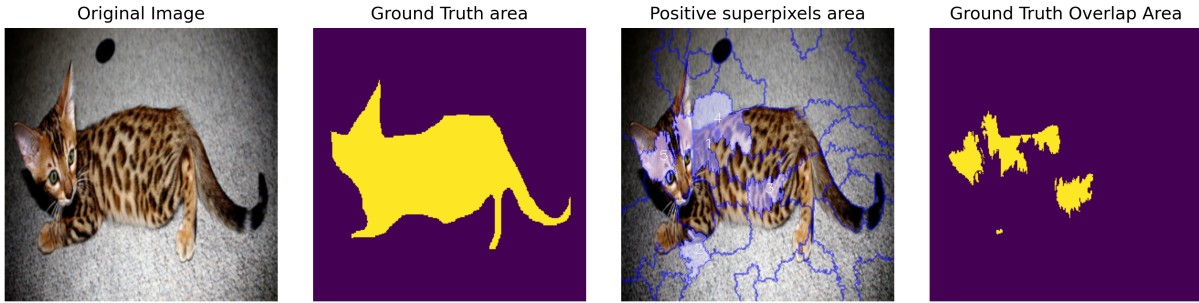

| Original Image | Ground Truth area | Positive superpixels area | Ground Truth Overlap Area |

Figure 3: Ground Truth Overlap (GTO) metric. The intersection between the top-k positive superpixels and the segmentation map is computed and normalized by the area of top-k positive superpixels. In this example, GTO = 0.58.

## 5  Results

We have performed extensive experimentation to test the claims made by Bora et al. (2024). Our evaluation focuses on the interpretability, consistency, and fidelity improvements claimed for SLICE over LIME. We also introduce GRID-LIME as an alternative method and assess its effectiveness.

In this section, we provide a detailed analysis of the consistency and fidelity of explanations produced by these methods. We examine the impact of Sign Entropy-based Feature Elimination (SEFE) and Adaptive Gaussian Blur, and conduct an ablation study to determine their individual contributions. Additionally, we extend our analysis beyond the original paper by introducing the Ground Truth Overlap (GTO) metric to compare explanation maps with segmentation labels.

Overall, our results partially support the claims of Bora et al. (2024). While we confirm that SLICE produces significantly more consistent explanations than LIME, we find that its fidelity improvements are less conclusive. Additionally, we identify potential biases in the evaluation methodology, particularly in the

---

[4]https://github.com/CSBXAI/slice_reproducibility

choice of perturbation strategies. Our findings suggest that while SLICE improves consistency, further work is required to ensure fair fidelity comparisons.

Furthermore, we provide additional qualitative examples in Appendix F, illustrating the differences in explanation maps generated by LIME, SLICE, and GRID-LIME. These visualizations help to better understand the distinct attribution patterns produced by each method. In Appendix B, we also present an alternative AOPC evaluation where SLICE is tested under the same perturbation conditions as LIME and GRID-LIME, leading to more comparable fidelity results.

In the following subsections, we analyze each claim in detail and present supporting evidence.

### 5.1 Claim 1: LIME suffers from inconsistencies due to sign flip and rank variance

To assess explanation stability, we compare the Combined Consistency Metric (CCM) distributions across methods. Figure 4 presents the Kernel Density Estimation (KDE) plots of CCM scores for SLICE, LIME, and GRID-LIME across different datasets and models.

The results indicate that SLICE produces significantly more stable explanations than LIME, as evidenced by a strong peak near 1.0, meaning that sign flips and ranking inconsistencies are minimized. In contrast, LIME's CCM values are more widely spread, confirming instability. GRID-LIME exhibits a CCM distribution between that of LIME and SLICE, indicating improved stability while maintaining flexibility.

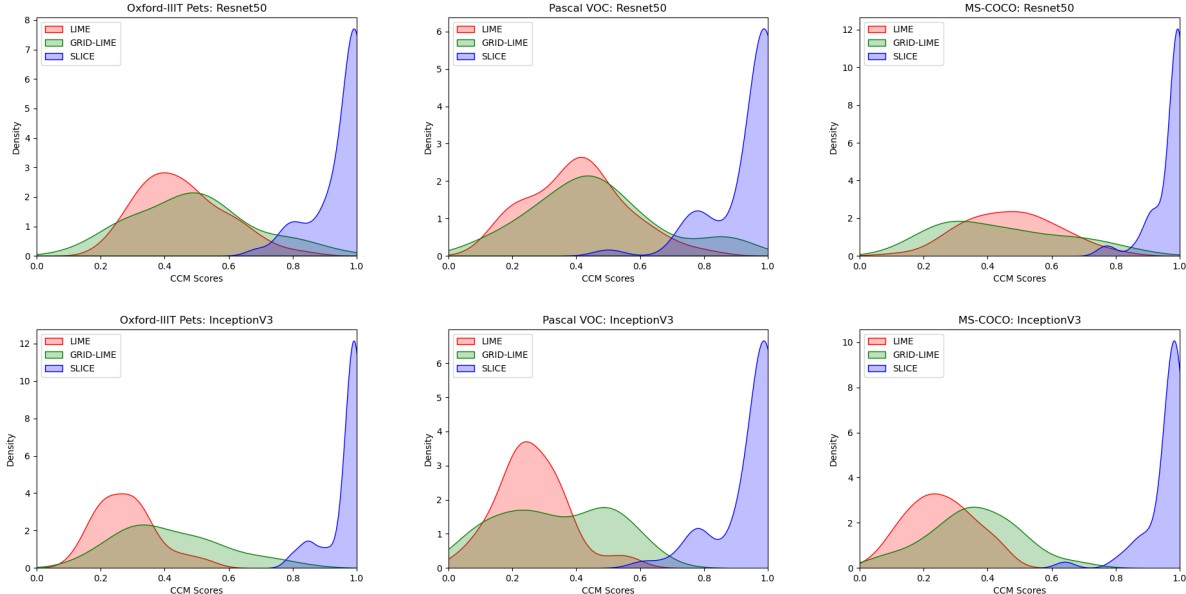

Figure 4: Kernel Density Estimation (KDE) plots of Combined Consistency Metric (CCM) scores for SLICE, LIME, and GRID-LIME. A higher peak in CCM indicates greater explanation consistency, while a wider spread reflects higher variability.

### 5.2 Claim 2: SEFE improves stability in explanations

The impact of Sign Entropy-based Feature Elimination (SEFE) is evaluated using the Combined Consistency Metric (CCM), which measures explanation stability. Figure 5 consists of six plots, each containing three KDE curves representing different methods. Across these plots, SLICE consistently achieves the highest CCM, with values closest to 1, confirming that it provides the most stable explanations. SLICE_BLUR follows with a CCM around 0.9, indicating strong stability. In contrast, SLICE_SEFE alone exhibits much lower stability, with a CCM ranging between 0.2 and 0.3. While SEFE reduces sign flips and ranking inconsistencies, its direct impact on stability is limited. The improvement in CCM is primarily observed

when SEFE is combined with Adaptive Gaussian Blur, rather than when used in isolation. These results suggest that SEFE alone is insufficient for improving stability in explanations and is most effective when integrated with Adaptive Gaussian Blur.

### 5.3 Claim 3: Adaptive Gaussian Blur enhances explanation consistency

To evaluate the effect of Adaptive Gaussian Blur, we compare the adjusted $R^2$ values of Ridge regression models trained with different $\sigma$ values. Our results show that dynamically selecting $\sigma$ based on entropy increases $R^2$, suggesting that perturbations remain realistic while improving consistency. However, the improvement is not always significant across all cases. This can also be observed in Figure 5, where Adaptive Gaussian Blur (AdaBlur) achieves relatively high CCM values, but the difference compared to SLICE without Adaptive Gaussian Blur is not always significant. These findings indicate that while Adaptive Gaussian Blur contributes to stability, its impact depends on the specific setting and dataset.

#### 5.3.1 Ablation Study

We conducted an ablation study to replicate the findings of (Bora et al., 2024), evaluating the individual contributions of SEFE and AdaBlur in SLICE. Since SLICE consists of these two key components, we performed the ablation study by isolating each module, as shown in Figure 5.

Our results align with the original ablation study in (Bora et al., 2024). SLICE achieves the highest consistency when both components are included, confirming their complementary roles. Among the individual modules, AdaBlur contributes more to consistency than SEFE. The absence of AdaBlur leads to perturbed images that deviate further from the original, reducing consistency. These findings highlight the importance of using both components together for optimal stability.

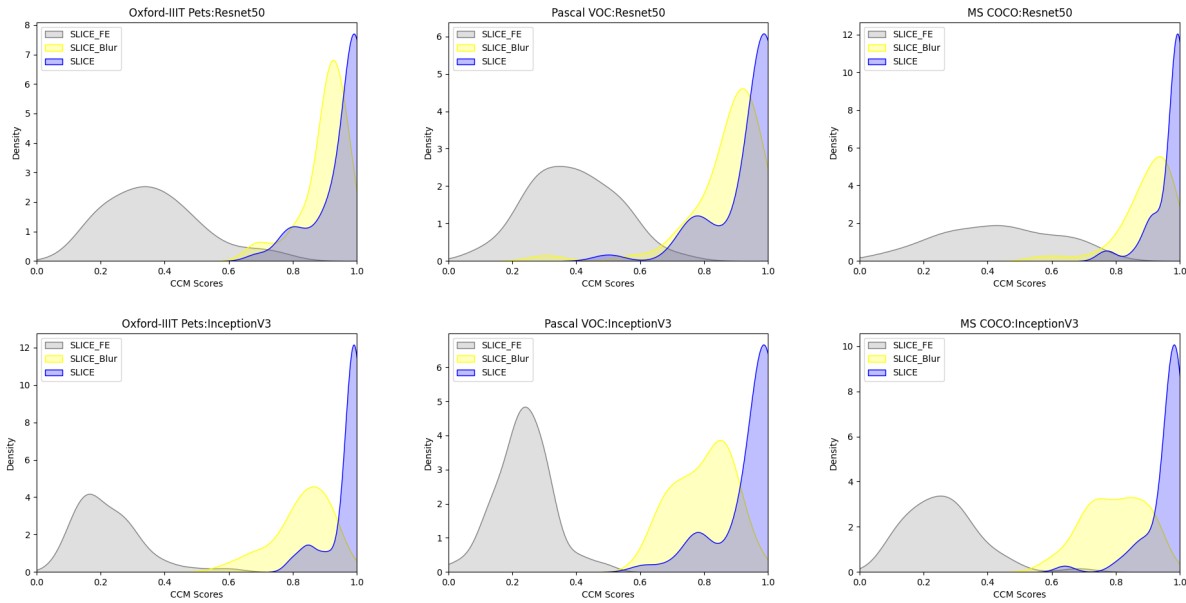

Figure 5: Distribution of CCM scores for ablation experiment. SLICE_FE shows the results for the SEFE while SLICE_Blur shows the results for the Adaptive Gaussian Blur component. SLICE shows the results for both components included. (higher is better).

### 5.4 Claim 4: SLICE provides better fidelity than LIME

To assess fidelity, we use the Area Over Perturbation Curve (AOPC) and Area Under Curve (AUC) metrics. Figure 6 presents the ECDF plots of AOPC scores. We performed the Most Relevant First (MoRF) procedure

similar to the author's where insertion refers to the insertion of most positive superpixels and deletion refers to the deletion of the most negative superpixels.

In the insertion task (first row of Figure 6), SLICE consistently achieves higher AOPC scores than LIME and GRID-LIME, demonstrating better preservation of important features. However, in the deletion task (second row), SLICE performs worse than both methods, suggesting its deletion strategy is less effective.

Figure 7 presents ECDF plots of AUC scores, where SLICE outperforms the other methods in both insertion (first row) and deletion (second row). A higher AUC value is preferable for insertion, while a lower AUC value is better for deletion, confirming SLICE's strong performance in both tasks. To further evaluate whether SLICE significantly differs from LIME and GRID-LIME across datasets we performed a Wilcoxon rank test on AOPC and AUC (Appendix A).

One potential reason for SLICE's mixed fidelity performance is the difference in perturbation strategies. SLICE applies Gaussian blur-based perturbations, whereas LIME and GRID-LIME use randomized occlusions. This methodological difference affects how feature importance is evaluated, potentially biasing AOPC and AUC scores. To address this, we introduce an alternative AOPC evaluation (Appendix B), where SLICE is tested under the same perturbation method as LIME and GRID-LIME. This ensures a fair comparison of fidelity by eliminating potential biases introduced by differing perturbation techniques.

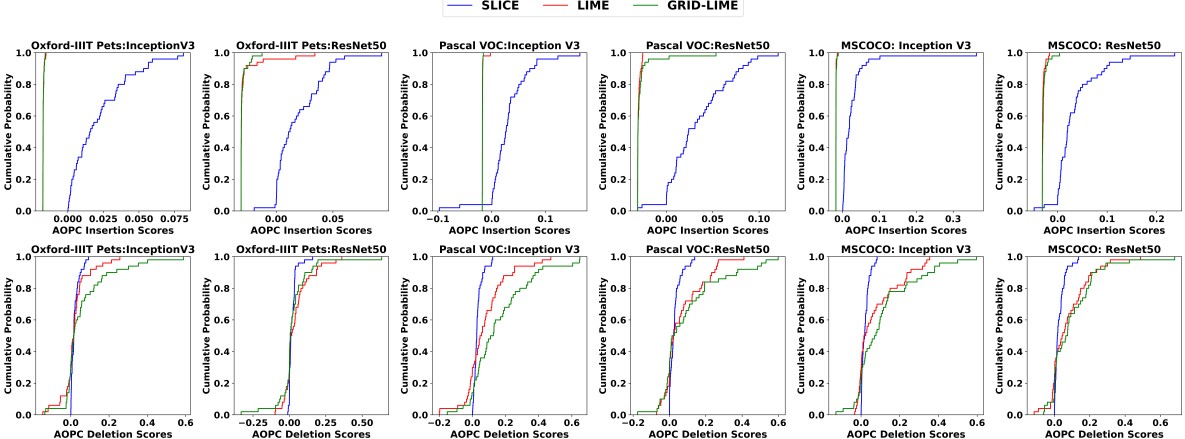

Figure 6: Empirical Cumulative Distribution Function (ECDF) plot of AOPC scores for different methods. The first row corresponds to the insertion task and the second row corresponds to the deletion task. A higher AOPC score indicates better fidelity in capturing important features.

## 5.5   Results Beyond the Original Paper

Beyond reproducing the original results, we introduce the Ground Truth Overlap (GTO) metric to evaluate how well explanations align with segmentation maps. Figure 8 presents the results.

LIME consistently achieves the highest GTO scores, indicating better alignment with ground truth segmentations. SLICE generally performs worse across most cases, while GRID-LIME performs better in about half the cases but remains further from LIME. This trend is expected since GTO is based on the Intersection over Union (IoU) of segmentation masks, and grid-based methods naturally achieve higher overlap due to their grid boxes.

Additionally, we introduce an alternative variation of the AOPC ECDF plot in Appendix B, where the insertion logic is modified. Unlike the standard evaluation, this version uses the perturbation method of LIME and GRID-LIME for SLICE instead of applying Gaussian blur. This adjustment ensures that SLICE is evaluated under the same perturbation conditions as the other methods, removing potential biases introduced by differing perturbation strategies. As a result, this version provides a more direct and fair comparison of

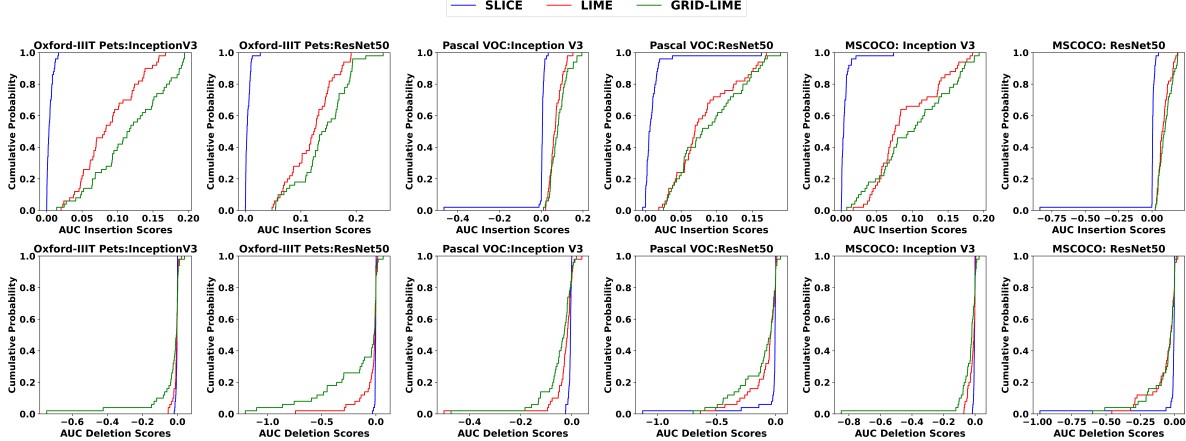

Figure 7: Empirical Cumulative Distribution Function (ECDF) plot of AUC scores for insertion (row 1) and deletion (row 2) tasks. For insertion, a higher AUC value is preferable, indicating that adding back important features improves model confidence. For deletion, a lower AUC value is better, as it suggests that removing key features leads to a significant drop in confidence.

fidelity. Under this setting, SLICE performs similarly to LIME and GRID-LIME for insertion but slightly worse for deletion.

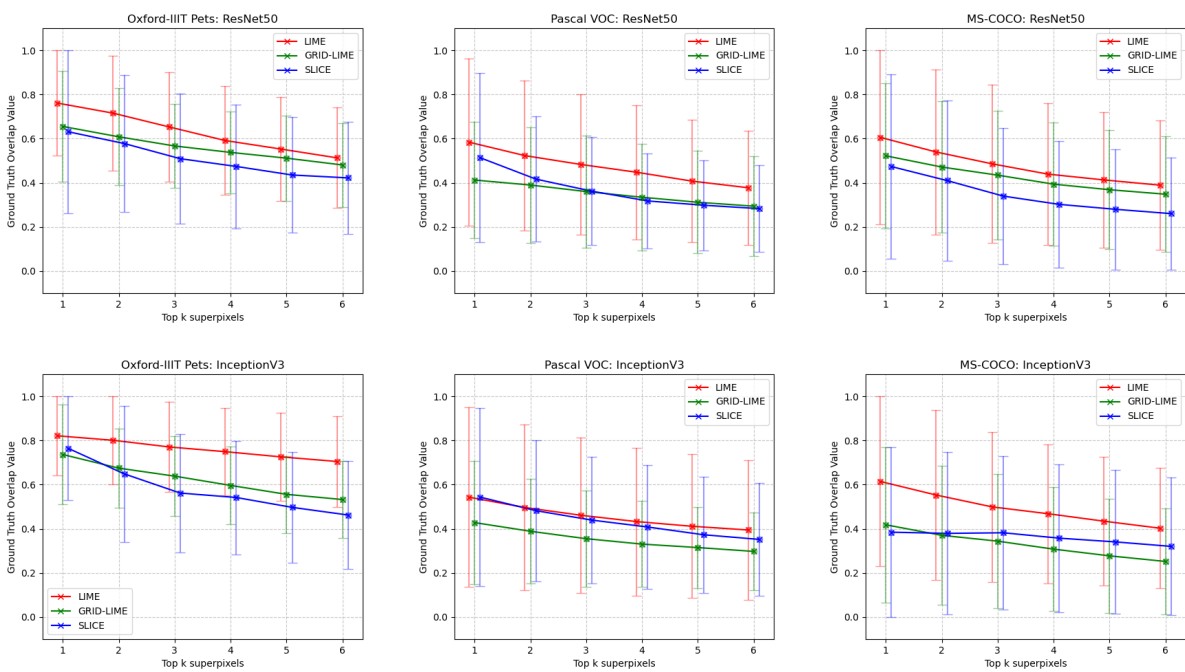

Figure 8: Ground Truth Overlap (GTO) plots, where the x-axis represents the number of top-k ranked superpixels ranging from 1 to 6, and the y-axis indicates the GTO score (higher is better). LIME achieves the highest alignment with segmentation maps, GRID-LIME performs moderately well, and SLICE lags behind in most cases.

## 6 Discussion

Our results partially support the claims made by Bora et al. (2024), confirming that SLICE improves explanation consistency but challenging the assertion that it provides superior fidelity. Additionally, we identified methodological limitations in the original study, particularly in how perturbation strategies affect fidelity evaluations.

Across the Oxford-IIIT Pets, PASCAL VOC, and MS COCO datasets, SLICE consistently produced more stable explanations than LIME, as reflected in its higher Combined Consistency Metric (CCM) scores. However, our fidelity analysis using AOPC and AUC showed that SLICE did not always outperform LIME. Our newly introduced Ground Truth Overlap (GTO) metric further suggested that LIME often produces more segmentation-aligned attributions, despite its instability. This discrepancy likely arises from SLICE's use of Gaussian blur-based perturbations, which affect how feature importance is preserved compared to LIME's direct occlusions. To address this, we introduced an alternative AOPC evaluation (Appendix B) where SLICE was tested using the same perturbation strategy as LIME and GRID-LIME, ensuring a fairer fidelity comparison. Under these conditions, SLICE performed similarly to the other methods for insertion but slightly worse for deletion, suggesting that its fidelity advantage in the original setup may be influenced by differences in perturbation techniques rather than intrinsic method effectiveness.

Another important limitation of SLICE is its computational cost. Our runtime analysis revealed that SLICE is significantly slower than LIME, requiring up to 13 times more processing time per experiment. While SLICE improves explanation stability, its increased computational complexity and energy consumption make it less practical for large-scale applications. As an alternative, we introduced GRID-LIME, which replaces LIME's segmentation-based superpixels with a structured grid. Our results showed that GRID-LIME improves explanation stability while maintaining a computational cost closer to LIME than SLICE as shown in Table 4 in Appendix D. This suggests that part of LIME's inconsistency stems from its segmentation process rather than its perturbation mechanism. However, GRID-LIME's rigid grid-based approach does not always align well with object boundaries, potentially affecting interpretability in certain cases.

A key limitation of our study is that our implementation of SLICE may differ slightly from the authors' original setup. While we referenced their official SLICE code, certain hyperparameter choices and evaluation metric implementations (such as ASFE, ARS, and CCM) had to be re-implemented based on the paper's descriptions. Despite these challenges, our extensive experimentation and sensitivity analysis suggest that our findings remain robust.

Overall, our study highlights trade-offs in post-hoc explainability methods. While SLICE improves consistency, its computational cost and fidelity claims require further scrutiny. Future research should focus on refining perturbation techniques to overcome the shift in distribution of input image, optimizing computational efficiency, and developing alternative fidelity metrics that do not rely solely on perturbation-based evaluations.

### 6.1 What was easy

The authors provided a GitHub repository containing an implementation of SLICE, which facilitated the reproduction of initial results. We were able to validate the consistency of sign and ranking of important superpixels for two images using their provided pickle files. Communication with the authors was helpful, as they provided missing metric code for validation upon request.

### 6.2 What was difficult

Several challenges emerged during reproduction. The codebase was incomplete, lacking implementations for key metrics such as ASFE, ARS, and CCM. Additionally, AOPC scores were not initially included. While the authors later provided an implementation for AOPC, the other metrics had to be inferred based on the paper's descriptions.

Another major limitation was computational efficiency. Running the original SLICE implementation which was in Tensorflow took approximately 15 minutes per image, making large-scale experiments infeasible. We

improved runtime by rewriting the implementation in PyTorch and incorporating multi-threading, which enabled us to conduct more comprehensive evaluations.

### 6.3 Communication with original authors

After reviewing the GitHub repository, we identified missing components and contacted the authors for clarification. They responded positively and provided the missing code for AOPC calculations. However, ASFE, ARS, and CCM metric implementations were never made available, requiring us to re-implement them independently. Despite this, the provided assistance facilitated certain aspects of reproduction.

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

## A  Wilcoxon rank test

To confirm significance, we conducted a Wilcoxon rank test on AOPC (Table 1) and AUC (Table 2), where D:M denotes Dataset:Model. O represents Oxford-IIT Pets, P refers to Pascal VOC, C refers to MS COCO, I indicates InceptionV3, and R denotes ResNet50. W stands for the Wilcoxon signed rank statistic, p-value is the probability of observing the data under the null hypothesis that the median difference is zero, $M_\Delta$ denotes the median difference of the pair-wise scores, and N.C refers to the number of negative differences (out of 50 images). While SLICE significantly outperforms LIME in AOPC-Insertion, the AOPC-Deletion and the AUC test do not show consistent differences across all comparisons.

Table 1: Wilcoxon rank test results for comparison of LIME(L), SLICE(S), and GRID-LIME(G) on AOPC. AOPC(x,y) tests whether the median difference AOPCscore(x) - AOPCscore(y) is zero.

| Test | D:M | W | p-value | $M_\Delta$ | N.C |
|---|---|---|---|---|---|
| **Insertion** | | | | | |
| AOPC(S,L) | O:I | 1275 | 8.88e-16 | 0.03286 | 0 |
| AOPC(S,L) | O:R | 1267 | 2.22e-14 | 0.04166 | 2 |
| AOPC(S,L) | P:I | 1209 | 1.74e-10 | 0.04280 | 2 |
| AOPC(S,L) | P:R | 1274 | 1.78e-15 | 0.05409 | 1 |
| AOPC(S,L) | C:I | 1275 | 8.88e-16 | 0.03483 | 0 |
| AOPC(S,L) | C:R | 1272 | 4.44e-15 | 0.04970 | 2 |
| AOPC(S,G) | O:I | 1275 | 8.88e-16 | 0.03255 | 0 |
| AOPC(S,G) | O:R | 1275 | 8.88e-16 | 0.04088 | 0 |
| AOPC(S,G) | P:I | 1210 | 1.57e-10 | 0.04353 | 2 |
| AOPC(S,G) | P:R | 1255 | 3.30e-13 | 0.05474 | 2 |
| AOPC(S,G) | C:I | 1275 | 8.88e-16 | 0.03405 | 0 |
| AOPC(S,G) | C:R | 1246 | 1.54e-12 | 0.05021 | 1 |
| **Deletion** | | | | | |
| AOPC(S,L) | O:I | 727 | 1.97e-01 | 0.00593 | 20 |
| AOPC(S,L) | O:R | 483 | 9.32e-01 | -0.01037 | 29 |
| AOPC(S,L) | P:I | 424 | 9.81e-01 | -0.02597 | 29 |
| AOPC(S,L) | P:R | 507 | 8.96e-01 | -0.00089 | 26 |
| AOPC(S,L) | C:I | 412 | 9.86e-01 | -0.00477 | 27 |
| AOPC(S,L) | C:R | 414 | 9.85e-01 | -0.01545 | 29 |
| AOPC(S,G) | O:I | 527 | 8.57e-01 | -0.00324 | 26 |
| AOPC(S,G) | O:R | 694 | 2.96e-01 | 0.00505 | 21 |
| AOPC(S,G) | P:I | 206 | 1.00e+00 | -0.08023 | 36 |
| AOPC(S,G) | P:R | 457 | 9.60e-01 | -0.00174 | 25 |
| AOPC(S,G) | C:I | 281 | 1.00e+00 | -0.02752 | 34 |
| AOPC(S,G) | C:R | 308 | 9.99e-01 | -0.03289 | 34 |

Table 2: Wilcoxon rank test results for comparison of LIME(L), SLICE(S), and GRID-LIME(G) on AUC. AUC(x,y) tests whether the median difference in AUC scores is zero.

| Test | D:M | W | p-value | $M_\Delta$ | N.C |
|---|---|---|---|---|---|
| **Insertion** | | | | | |
| AUC(S,L) | O:I | 0 | 1.00e+00 | -0.07546 | 50 |
| AUC(S,L) | O:R | 0 | 1.00e+00 | -0.11903 | 50 |
| AUC(S,L) | P:I | 0 | 1.00e+00 | -0.05818 | 50 |
| AUC(S,L) | P:R | 41 | 1.00e+00 | -0.06117 | 49 |
| AUC(S,L) | C:I | 0 | 1.00e+00 | -0.06956 | 50 |
| AUC(S,L) | C:R | 0 | 1.00e+00 | -0.06839 | 50 |
| AUC(S,G) | O:I | 0 | 1.00e+00 | -0.11019 | 50 |
| AUC(S,G) | O:R | 0 | 1.00e+00 | -0.13420 | 50 |
| AUC(S,G) | P:I | 0 | 1.00e+00 | -0.07070 | 50 |
| AUC(S,G) | P:R | 27 | 1.00e+00 | -0.06947 | 49 |
| AUC(S,G) | C:I | 0 | 1.00e+00 | -0.09347 | 50 |
| AUC(S,G) | C:R | 1 | 1.00e+00 | -0.08312 | 49 |
| **Deletion** | | | | | |
| AUC(S,L) | O:I | 729 | 1.91e-01 | 3.27e-05 | 25 |
| AUC(S,L) | O:R | 989 | 2.44e-04 | 0.00554 | 16 |
| AUC(S,L) | P:I | 1079 | 3.36e-06 | 0.01253 | 14 |
| AUC(S,L) | P:R | 1064 | 7.60e-06 | 0.03557 | 15 |
| AUC(S,L) | C:I | 936 | 1.71e-03 | 0.00466 | 22 |
| AUC(S,L) | C:R | 1051 | 1.49e-05 | 0.01683 | 13 |
| AUC(S,G) | O:I | 907 | 4.29e-03 | 0.00284 | 20 |
| AUC(S,G) | O:R | 1016 | 7.79e-05 | 0.01112 | 15 |
| AUC(S,G) | P:I | 1144 | 5.25e-08 | 0.02479 | 12 |
| AUC(S,G) | P:R | 1049 | 1.64e-05 | 0.03463 | 13 |
| AUC(S,G) | C:I | 1087 | 2.13e-06 | 0.01570 | 13 |
| AUC(S,G) | C:R | 1060 | 9.38e-06 | 0.02105 | 13 |

# B  Alternative AOPC Scores

A modified version of the ECDF plot of AOPC scores with $\sigma = 0$ is presented in Figure 9. In this version, $\sigma = 0$ denotes that SLICE does not apply Gaussian blurring; rather, it uses the same perturbation strategy as LIME and GRID-LIME, masking selected superpixels by setting them to zero. This ensures a fairer evaluation by placing SLICE on the same evaluation framework as the other methods.

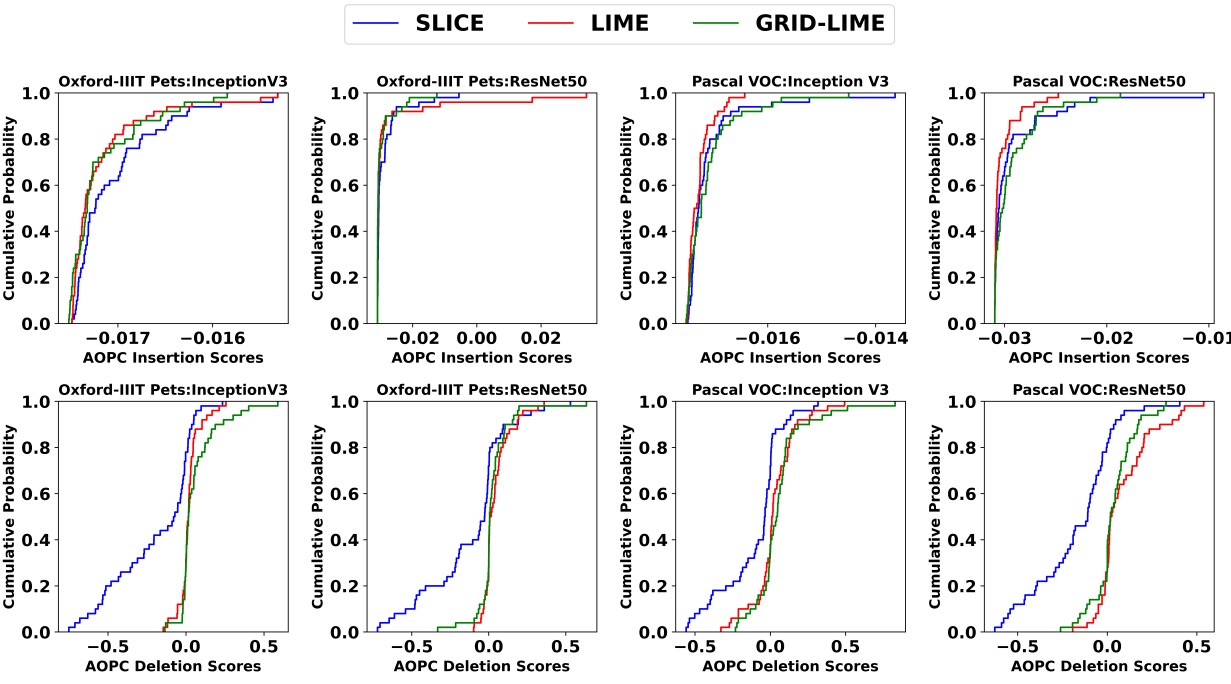

Figure 9: Alternative ECDF plot of AOPC scores with $\sigma = 0$, using LIME and GRID-LIME's perturbation method for SLICE instead of Gaussian blur (higher is better).

# C  Hyperparameters

Table 3: Hyperparameter values used in our experiments for SLICE, LIME, and GRID-LIME. The values were determined through empirical validation and small-scale hyperparameter searches to ensure fair and reproducible comparisons. For SLICE, we optimized the Adaptive Gaussian Blur $\sigma$ using an iterative search strategy, while GRID-LIME's grid box size is determined dynamically based on perturbation consistency.

| Hyperparameter | Value |
|---|---|
| Adaptive Gaussian Blur $\sigma$ search range | {0.1, 0.2, 0.3, 0.4, 0.5} |
| Number of Ridge Regression Models ($M$) | 1000 |
| Number of perturbations per image ($N$) | 500 |
| SEFE Tolerance | 3 |
| Number of iterations per image | 10 |
| Number of images sampled per dataset | 50 |
| Quickshift kernel size | 5 |
| Quickshift max distance | 200 |
| Quickshift ratio | 0.2 |

## D   Computational requirements and environmental impact

The table below presents an analysis of computational runtime and associated carbon emissions for different model-explainer-dataset combinations. The explainers considered include LIME, SLICE, and GRID-LIME, applied to ResNet50 and InceptionV3 models trained on three datasets: Oxford-IIIT Pets, Pascal VOC, and MS COCO. The objective is to assess the environmental impact of these machine learning experiments in terms of both execution time and carbon emissions.

The values in the table were estimated using the `ML CO`$_2$ `Impact calculator`[5], a tool that provides carbon emission estimates based on hardware type, runtime, and cloud provider. One NVIDIA A100 GPU was used for all computations. The estimated carbon footprint for each experiment is measured in kilograms of $CO_2$ equivalent.

The `ML CO`$_2$ `Impact calculator` determines emissions based on multiple factors, including hardware efficiency, runtime, and the energy grid's carbon intensity. The results indicate clear differences between the explainability methods. LIME and GRID-LIME have consistently lower runtime and emissions compared to SLICE. InceptionV3 requires more computational resources than ResNet50, leading to increased carbon emissions. Dataset size also influences results, with larger datasets requiring more processing time and energy.

Table 4: Computational runtime and carbon emissions per experiment.

| Explainer | Model | Dataset | Time (min) | Carbon emissions (kg $CO_2$e) |
|-----------|-------|---------|------------|-------------------------------|
| LIME | ResNet50 | Oxford-IIIT Pets | 34 | 0.05 |
| LIME | ResNet50 | Pascal VOC | 35 | 0.05 |
| LIME | ResNet50 | MSCOCO | 37 | 0.06 |
| LIME | InceptionV3 | Oxford-IIIT Pets | 51 | 0.11 |
| LIME | InceptionV3 | Pascal VOC | 52 | 0.11 |
| LIME | InceptionV3 | MS COCO | 52 | 0.11 |
| SLICE | ResNet50 | Oxford-IIIT Pets | 332 | 0.65 |
| SLICE | ResNet50 | Pascal VOC | 328 | 0.65 |
| SLICE | ResNet50 | MS COCO | 334 | 0.65 |
| SLICE | InceptionV3 | Oxford-IIIT Pets | 576 | 1.08 |
| SLICE | InceptionV3 | Pascal VOC | 581 | 1.08 |
| SLICE | InceptionV3 | MS COCO | 593 | 1.08 |
| GRID-LIME | ResNet50 | Oxford-IIIT Pets | 35 | 0.05 |
| GRID-LIME | ResNet50 | Pascal VOC | 39 | 0.06 |
| GRID-LIME | ResNet50 | MS COCO | 41 | 0.07 |
| GRID-LIME | InceptionV3 | Oxford-IIIT Pets | 52 | 0.11 |
| GRID-LIME | InceptionV3 | Pascal VOC | 53 | 0.11 |
| GRID-LIME | InceptionV3 | MS COCO | 53 | 0.11 |

---

[5]`https://mlco2.github.io/impact/`

# E    Results from original paper

We reproduced the authors' Kernel Density Estimation (KDE) plots of the Combined Consistency Metric (CCM) scores for SLICE using their TensorFlow implementation, as shown in Figure 10.

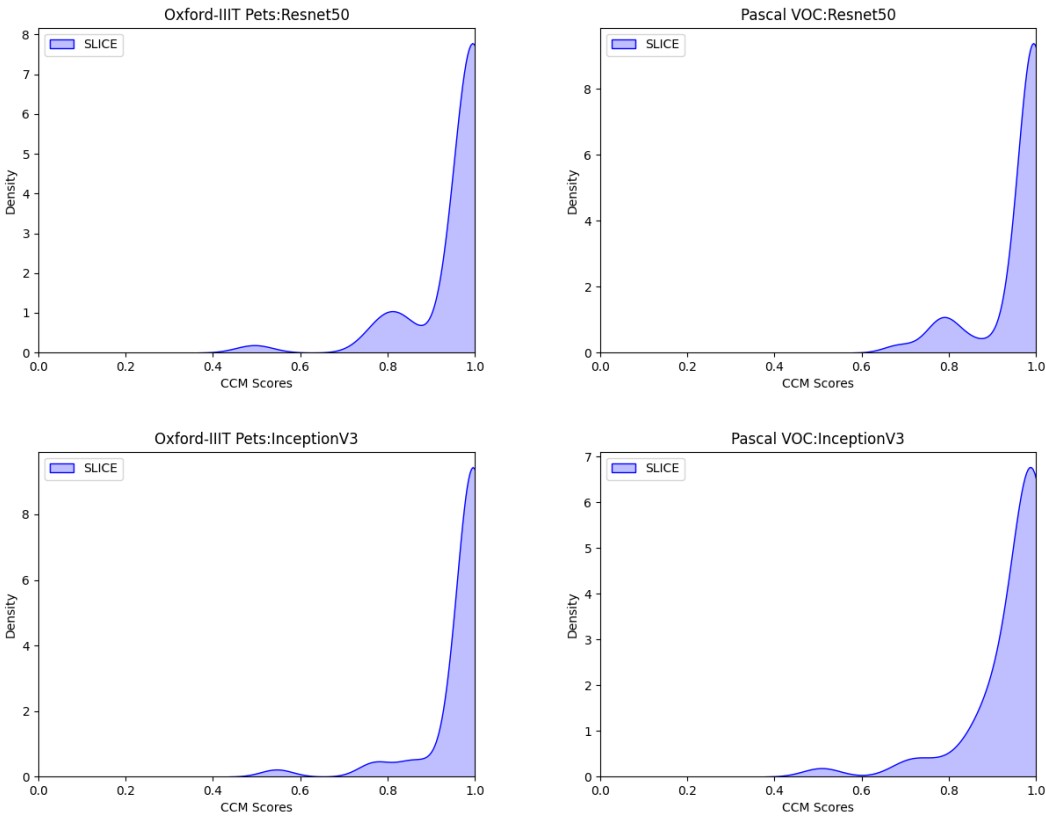

Figure 10: Kernel Density Estimation (KDE) plots of the Combined Consistency Metric (CCM) scores for SLICE using the author's code.

## F    Qualitative Examples

To provide additional insight into the explanations generated by each method, we present qualitative examples of LIME, SLICE, and GRID-LIME. These visualizations highlight the ranked positive and negative superpixels, showcasing how each explainer method attributes importance to different image regions.

### F.1    LIME Explanations

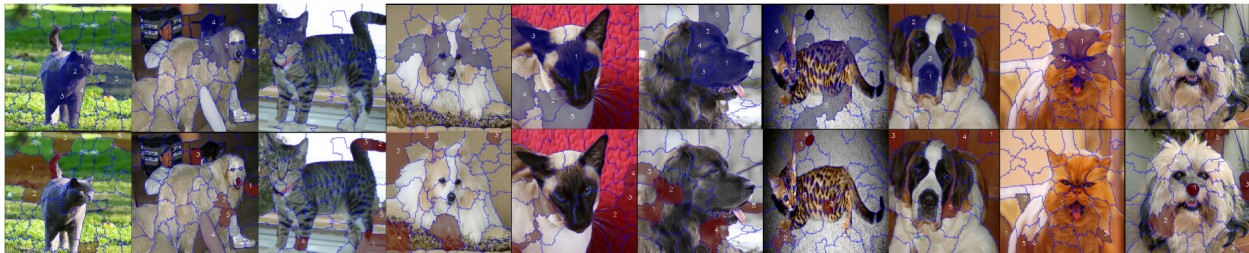

Figure 11: Qualitative examples of LIME explanations, displaying ranked positive (highlighted in warm colors) and negative (highlighted in cool colors) superpixels. LIME assigns importance by sampling perturbed versions of the image and fitting a linear model to approximate the model's decision boundary.

### F.2    SLICE Explanations

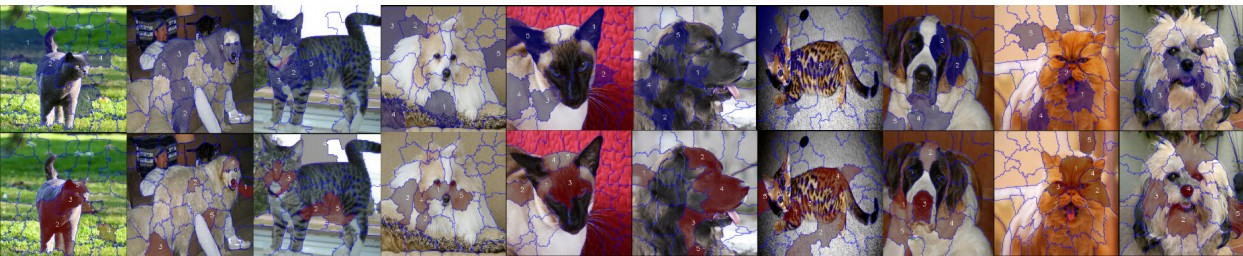

Figure 12: Qualitative examples of SLICE explanations, showing ranked positive and negative superpixels. SLICE applies spatially-aware perturbations and adaptive feature elimination, improving local consistency in attributions.

### F.3    GRID-LIME Explanations

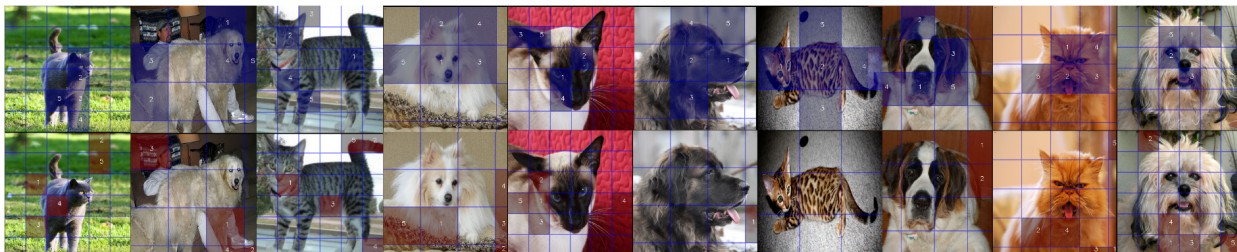

Figure 13: Qualitative examples of GRID-LIME explanations, where attributions are structured based on a predefined grid. Positive and negative superpixels are ranked similarly to LIME but constrained within fixed spatial regions, leading to different interpretations.

### F.4 Consistency of LIME, SLICE, and GRID-LIME

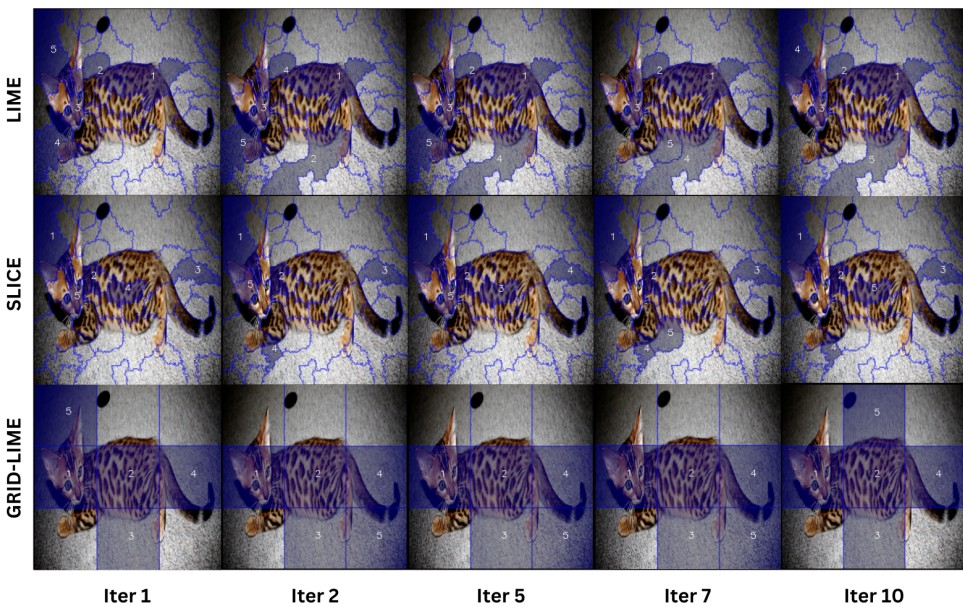

Figure 14: Examples of positive superpixels returned over multiple runs of LIME, SLICE, and GRID-LIME. LIME exhibits the highest variability, while SLICE demonstrates the most consistent attributions.

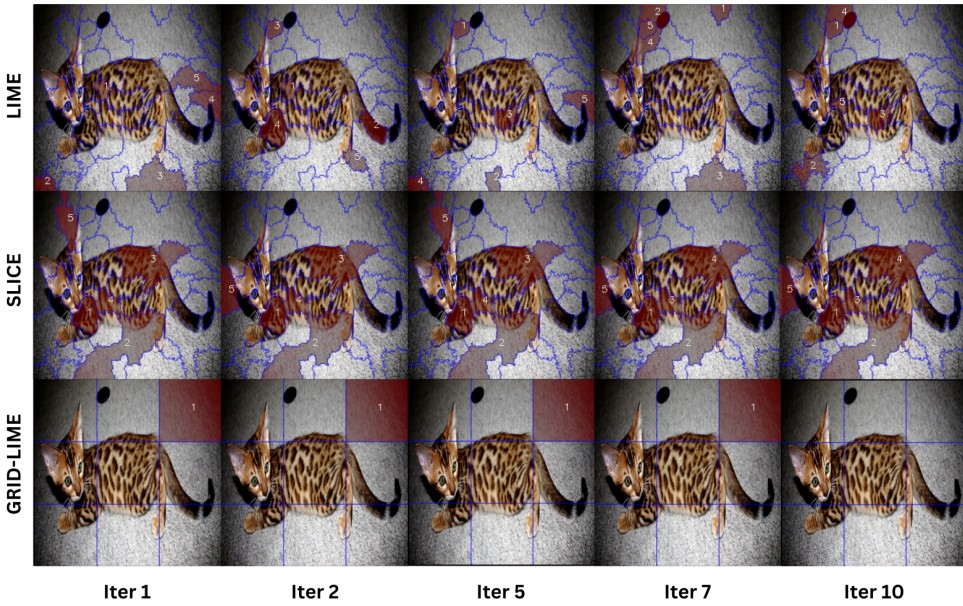

Figure 15: Examples of negative superpixels returned over multiple runs of LIME, SLICE, and GRID-LIME.

