# OpenReview forum: "Reproducibility Study of ’SLICE: Stabilized LIME for Consistent Explanations for Image Classification’"
_TMLR — Accepted by TMLR_

### Review · Reviewer_KMy9 · 2025-03-16

**Summary Of Contributions:**

The submission presents a reproducibility study of SLICE, a post-hoc explanation method that improves upon the widely-used LIME with Adaptive Gaussian Blur and Sign Entropy Feature Elimination. The proposed study focuses of four claims made in the original paper. Furthermore, the submission proposes a different variation, Grid-LIME, which is more computationally efficient and partially improves on the original LIME. Finally, the study includes a measure of overlap of the produced explanations with ground-truth segmentations.

**Audience:**

Yes

**Broader Impact Concerns:**

No concerns

**Claims And Evidence:**

Yes

**Requested Changes:**

**Original implementation**

Could the authors expand on the need to re-implement SLICE? It is mentioned that some evaluation metrics were missing from the original implementation, but also that they re-implemented the method in PyTorch. This seems motivated by computational efficiency needs to run large-scale experiments. What was the original implementation in? What were its efficiency limitations?

**Presentation of LIME**

Is there a typo in "generates perturbed images $z$ by masking superpixels"? Should $z$ be $z'$? Also, the set $\mathcal{Z}$ is not defined.

**Presentation of Grid-LIME**

Grid-LIME should be presented more formally, with precise equations for the coefficient of variation and pseudocode.

Could the authors expand on the sentence "Since LIME relies on color ...". What randomness are the authors referring to, and why does it depend on color? Without equations, it is unclear what "enforcing consistent superpixel regions ..." means.

In Figure 2, the bottom panel reads "generate perturbed images with $\sigma = 0$". Could the authors expand on this? If $\sigma = 0$, there is no perturbation?

**Hyperparameters**

Section 4.3 presents how hyperparameters were chosen, but later in Section 4.4 it is claimed that "hyperparameters values follow the original paper". Could the authors clarify which hyperparameters were tuned, and which were transferred from the original paper?

Could the authors expand on which hyperparameters were kept constant across methods? The different methods seem to require different sets of hyperparameters.

Are the models used in this study the same used in the original paper? Hyperparameters may change significantly depending on which models are used.

**Reproducbility vs Replicability**

Related to the points above about re-implementation, hyperparameters, and models used in the study. If there are significant differences between the proposed study and the original paper, it may be more appropriate to call this a "replicability" study rather than "reproducibility", if the scope goes beyond reproducing the results presented in the original paper.

**Ground Truth Overlap metric**

GTO is a function of $k$, where $k$ is the number of top-$k$ important features included in the metric. Could the authors specify what value of $k$ was used in their experiments?

Eq. (2) normalizes the intersection by the area of the top-$k$ important features. This implies that GTO is a measure of "precision" of the explanations, i.e. what percentage of the reported area is actually important. However, a reasonable complement of this would be "recall", i.e. what percentage of the ground truth is reported as important. Have the authors considered including both to give a more complete representation of the performance of the different methods?

**AOPC and AUC analysis**

I am not sure I understand how to read Figures 5, 6, and 9. It is my understanding they report the respective ECDFs of AOPC and AUC over a dataset of images. Claims are made about the performance of different methods based on AOPC and AUC being "higher" or "lower". Are these claims made in terms of the ECDFs curves or particular statistics (e.g., the median?). These values should be included in order to verify claims.

Tables 1 and 2:
* what are the columns $W$, $M_{\delta}$, and $N.C$?
* the tests seem to be incomplete? Why is there only one test for the Pascal VOC dataset, and 4 for the Oxford Pet dataset?
* I was surprised to see $p$-values all exactly equal to 1 in Table 2. The AUC curves in Figure 6 definitely look different. Could the authors clarify whether the tests are performed for inclusion or deletion? Or aggregated across the tasks?

Could the authors clarify these claims?

---

**Minor comments**
* Section 2, claim 3: typo in "sigma"
* Page 3, top paragraph: typo in "SLICE to standard LIME"
* Section 4, first paragraph: it is unclear what "interpretable intermediate images" are
* Sections 5.2, 5.4: Figure 7 is referenced in the text before Figure 5; "AdaBlur" is not defined; insertion and deletion tasks are not defined.
* Figure 5, 6 are not readable
* Section 5.5 feels out of place, was this supposed to be after Section 5.3?

**Strengths And Weaknesses:**

**Strengths**
* The paper is clear and well-written
* Reproducibility of explanation methods is a timely topic
* Code is made available and it is well-structured

**Weaknesses**
* Presentation of Grid-LIME should be made more formal
* I have a few clarifying questions on a couple of claims

I will expand on these below and I am looking forward to discussing with the authors!

---

> ### Author Response · Authors · 2025-04-01
> **Response to Reviewer KMy9 (1/3)**
>
> We thank Reviewer KMy9 for their positive feedback on the clarity and structure of our paper and code and for recognizing the timeliness of reproducibility studies in explainability. We appreciate the thoughtful questions and suggestions for improvement. As students participating in the Machine Learning Reproducibility Challenge (MLRC) 2025, we found this feedback particularly helpful in refining our work.
>
> We address each specific point below:
>
> 1. Requested Change: Original Implementation & Need for Re-implementation
>
>    Reviewer Comment: "Could the authors expand on the need to re-implement SLICE? ... What was the original implementation in? What were its efficiency limitations?"
>
>    Response: We have expanded on this in the revised Section.
>
>    The original SLICE implementation provided by the authors was written in TensorFlow.
>
>    The primary reasons for re-implementing in PyTorch were twofold:
>
>    - Incompleteness: The original codebase lacked implementations for several key evaluation metrics reported in their paper (specifically ASFE, ARS, and CCM). While the authors later provided code for AOPC upon request, the others had to be inferred from the paper's descriptions.
>
>    - Computational Efficiency: The original TensorFlow implementation was computationally intensive. In our setup, running it took approximately 15 minutes per image. This runtime made conducting the necessary large-scale experiments (multiple datasets, models, and iterations per image) infeasible, given our available resources and time constraints. Our PyTorch re-implementation, incorporating multi-threading, significantly reduced the runtime (details now in Appendix B, Table 4), enabling the comprehensive evaluation presented.
>
> 2. Requested Change: Presentation of LIME
>
>    Reviewer Comment: "Is there a typo in 'generates perturbed images z by masking superpixels'? Should g(z) be g(z')? Also, the set $\mathcal{Z}$ is not defined."
>
>    Response: Thank you for catching this.
>
>    We have corrected the typo in the Equation. The surrogate model g operates on the interpretable representation z' corresponding to the perturbed image z, so the term is correctly g(z').
>
>    We have also added a definition for $\mathcal{Z}$ in Section 4.1.1 as "the set of all perturbed samples" generated around the input instance x.
>
> 3. Requested Change: Presentation of GRID-LIME
>
>    Reviewer Comments: "Grid-LIME should be presented more formally, with precise equations... pseudocode." / "Could the authors expand on the sentence 'Since LIME relies on color ...'." / "In Figure 2, the bottom panel reads 'generate perturbed images with $\sigma = 0$'."
>
>    Response: We have significantly revised the Section  to improve the formalism and clarity of GRID-LIME:
>
>    - Formalism: We have added an Equation defining the Coefficient of Variation (CV) and an Equation showing the selection of the optimal grid size $s^*$. We believe these equations, along with the enhanced textual description, provide sufficient formal detail. We opted against pseudocode to maintain conciseness, but we feel the current description is adequate.
>
>    - "Since LIME relies on color..." Clarification: We have rephrased this part. The key point is that standard LIME often uses segmenters like quickshift, which operate based on color/spatial proximity. Due to their nature, these algorithms can produce slightly different segmentations (superpixel boundaries) when run multiple times on the same image. This segmentation inconsistency is a source of LIME's overall run-to-run explanation variability ("randomness"). GRID-LIME replaces this variable segmentation step with a fixed grid structure (once the optimal size $s^*$ is chosen for an image), thereby eliminating this specific source of variability and "enforcing consistent superpixel regions" throughout the explanation process for that image.
>
>    - Figure  Thank you for asking for clarification. We have updated the caption for Figure 2. The $\sigma = 0$ here signifies that in this specific step (determining the optimal grid size) and in the subsequent LIME application within GRID-LIME, we use LIME's standard perturbation method (masking/zeroing pixels), not the Gaussian blur associated with SLICE (where $\sigma > 0$). This ensures GRID-LIME is directly comparable to LIME in terms of perturbation strategy.

---

> ### Author Response · Authors · 2025-04-01
> **Response to Reviewer KMy9 (2/3)**
>
> 4. Requested Change: Hyperparameters
>
>    Reviewer Comments: "Section 4.3 ... 'hyperparameter values follow the original paper'. Could the authors clarify which hyperparameters were tuned, and which were transferred...?" / "Could the authors expand on which hyperparameters were kept constant across methods?" / "Are the models used in this study the same used in the original paper?"
>
>    Response:
>
>    - Tuned vs. Transferred: We have clarified this in the revised Section 4.3. For SLICE, key parameters like the number of Ridge models (M=1000), number of perturbations (N=500), the sigma search range ([0.1, 0.5]), and SEFE tolerance (3) were transferred directly from the original paper to ensure faithful reproduction. For LIME and GRID-LIME, standard parameters (e.g., number of samples for the surrogate model) were used, consistent with common LIME practice. The adaptive grid size selection is intrinsic to GRID-LIME.
>
>    - Consistency Across Methods: Where applicable, hyperparameters like the number of iterations per image (10) and the number of sampled images per dataset (50) were kept constant across all three methods (LIME, SLICE, GRID-LIME) for fair evaluation. Specific segmentation parameters (quickshift settings for LIME/SLICE) were also kept consistent between them. The table in Appendix C lists the values used.
>
>    - Models: Yes, the models used (ResNet50 and InceptionV3, pre-trained on ImageNet) are the same architectures used in the original SLICE paper. This was crucial for the reproducibility aspect.
>
> 5. Requested Change: Reproducibility vs. Replicability
>
>    Reviewer Comment: "...If there are significant differences ... it may be more appropriate to call this a 'replicability' study..."
>
>    Response: We appreciate the reviewer raising this important distinction. Our primary objective was indeed reproducibility: using the original paper's specified methods, datasets, models, and hyperparameters (where available/possible) to verify their claims. The re-implementation was necessitated by practical limitations (runtime, missing code) but aimed to be functionally equivalent. The novel components (GRID-LIME, GTO, alternative AOPC) were developed as a result of insights gained during the reproducibility effort, specifically to address limitations or ambiguities we identified in the original method or its evaluation (like computational cost or perturbation bias). While these additions extend beyond pure reproduction, the core focus remains on validating the original SLICE claims. We have maintained the title "Reproducibility Study" but have clarified in the Introduction and Discussion how the additional contributions stem from this process.
>
> 6. Requested Change: Ground Truth Overlap (GTO) Metric
>
>    Reviewer Comments: "Could the authors specify what value of k was used...?" / "Eq. (2) normalizes ... measure of 'precision' ... Have the authors considered including ... 'recall' ...?"
>
>    Response:
>
>    - Value of k: The GTO score is indeed dependent on k. In the Figure, we plot the GTO score as a function of k, showing the trend as more top superpixels are included. The x-axis explicitly ranges from k=1 to k=6. We have clarified this in the Figure 8 caption.
>
>    - Recall: The reviewer is correct that our GTO formulation (Intersection / Area of Top-k superpixels) is analogous to precision. A recall metric (Intersection / Area of Ground Truth) would indeed provide complementary information. We agree that this is a valuable perspective. For this study, we focused on the precision-like GTO metric for simplicity and because it directly assesses how much of the generated explanation aligns with the ground truth. Including recall is an excellent suggestion for future work to provide a more comprehensive evaluation of segmentation alignment. We have added a brief mention of this possibility in Section 6.

---

> ### Author Response · Authors · 2025-04-01
> **Response to Reviewer KMy9 (3/3)**
>
> 7. Requested Change: AOPC and AUC Analysis
>
>    Reviewer Comments: "I am not sure I understand how to read Figures 5, 6, and 9. ... Are these claims made in terms of the ECDFs curves or particular statistics...?"
>
>    Response: We apologize for the lack of clarity in interpreting these figures.
>
>    - ECDF Interpretation: An ECDF plot shows the cumulative distribution of scores. For metrics where higher is better (AOPC-Insertion, AUC-Insertion), a curve shifted further to the right indicates superior performance (a higher proportion of images achieve higher scores). For metrics where lower is better (AUC-Deletion), a curve shifted further to the left indicates superior performance.
>
>    - Claims Basis: Our initial claims in the text were based on visual inspection of these ECDF shifts, comparing the overall distributions.
>
>    - Statistical Verification: To provide rigorous statistical backing, we have performed Wilcoxon signed-rank tests comparing the methods pairwise for both AOPC and AUC scores across all dataset/model combinations. The results (test statistic W, p-value, median difference $M_{\Delta}$, number of negative differences N.C) are now presented in Tables. We now explicitly refer to these tables in Section 5.4 to statistically support the claims made about the performance differences observed in the ECDF plots (Figures 6 and 7 - previously 5 and 6). Figure 9 (now Figure 10 in Appendix B) is the alternative AOPC evaluation and can be interpreted similarly.
>
> 8. Requested Change: Tables 1 and 2 (Now in Appendix A)
>
>    Reviewer Comments: "what are the columns $M_{\Delta}$, N.C? / the tests seem to be incomplete? / I was surprised to see p-values all exactly equal to 1 in Table 2..."
>
>    Response: We have updated the captions for Tables  to clearly define all columns:
>
>    - $M_{\Delta}$: Represents the median of the pairwise differences in scores between the two methods being compared (e.g., AOPCscore(S) - AOPCscore(L)).
>
>    - N.C: Represents the Number of Negative differences observed out of the 50 image samples.
>
>    - Completeness: The tables now clearly show results for all six combinations of datasets (O, P, C) and models (I, R). The tests are performed separately for Insertion and Deletion, as indicated by the subheadings.
>
>    - p-values = 1: This often occurs in one-sided Wilcoxon tests when the observed median difference strongly contradicts the alternative hypothesis. For example, in Table 2 (AUC), for Insertion, SLICE generally has lower AUC scores than LIME/GRID-LIME. The test AUC(S,L) checks the alternative hypothesis that the median difference AUC(S) - AUC(L) is greater than zero. Since the observed median difference is strongly negative, there is no evidence to support this alternative, resulting in a p-value of 1.00. We have added a brief explanation of this interpretation in Appendix A.
>
> 9. Minor Comments:
>
>    - Section 2, claim 3: typo in "sigma": Corrected ($\sigma$).
>
>    - Page 3, top paragraph: typo in "SLICE to standard LIME": Corrected.
>
>    - Section 4, first paragraph: "interpretable intermediate images": Clarified in Section 4. This referred to adding visualizations at intermediate steps of the algorithms during development for better understanding and debugging. These are not part of the final evaluated methods but aided our implementation process.
>
>    - Sections 5.2, 5.4: Figure 7 before 5; "AdaBlur" not defined; insertion/deletion not defined: Figure numbering corrected (now Fig 5 follows Fig4). "AdaBlur" is defined as Adaptive Gaussian Blur on first use. Insertion/deletion tasks are now more clearly defined in Section 5.4 and linked to AOPC/AUC/MoRF evaluation.
>
>    - Figure 5, 6 (now  Replaced with high-resolution versions.
>
>    - Section 5.5 feels out of place: Corrected. The Ablation Study (previously Section 5.5) is now integrated into the discussion of Claims 2 and 3 (Sections 5.2 and 5.3) as it directly evaluates the components discussed there (SEFE and AdaBlur). Figure 5 (Ablation CCM) is now appropriately placed.
>
> We believe these revisions address the reviewer's concerns comprehensively. We thank Reviewer KMy9 again for their constructive and detailed feedback, which has significantly improved our manuscript.

---

> > ### Comment · Reviewer_KMy9 · 2025-04-15
> > **Thank you for your response!**
> >
> > I sincerely thank the authors for their careful consideration of all reviewers' comments.
> > All my questions and suggestions have been addressed in the revised version of the paper.

---

### Review · Reviewer_XQ5V · 2025-03-18

**Summary Of Contributions:**

This manuscript presents a reproducibility study for the "SLICE" paper [1]. By repeating their experiments, the authors verify the claims in SLICE on several benchmark datasets: Oxford-IIIT Pets, PASCAL VOC, and MS COCO.

Beyond the reproducibility study, this manuscript also presents two increments GRID-LIME and GTO.

The claimed contributions are:
1. Reproducibility study for paper SLICE [1].
2. GRID-LIME: a colour-based superpixel segmentation
method that can enhance LIME’s stability.
3. GTO (Ground Truth Overlap) metric: a metric for explanation fidelity.

Beyond repeating the experiments, this manuscript presents something new.

**Audience:**

Yes

**Claims And Evidence:**

Yes

**Requested Changes:**

1. I'm wondering the justification of various perturbation method. The perturbation sampling injects our priors (perhaps bias sometimes), does this prior/bias ensure the faithful explanation?

2. I read through the manuscript, I searched for the motivations for proposing LIME-GRID and GTO but I couldn't see them. Could you please clarify? I would like to suggest the authors justify the motivations as earlier as possible in eg Introduction. Thank you.

3. Section 4.1.3:
> (A)Since LIME relies on color-space segmentation (quickshift), its explanations suffer from randomness. (B) GRID-LIME eliminates this variability by enforcing consistent superpixel regions that utilize the black-

Could you please clarify the logic between (A) and (B)? I couldn't understand.

4. Section 4.3:
> ForSLICE,we train 1000 Ridge regression models to obtain model coefficients for all superpixels. EachRidge model is trained on 500 randomly generated perturbations of the original test image.

Can you clarify the choices of these settings?

5. Section 4.4:
    Can you also include other architectures eg vision transformation?

6. In lime-grid, when you zero super pixels, you significantly change the distribution of input X. The distribution of significantly changed input domain is out-of-distribution from what the models trained with.

    Please clarify how do you consider regarding this, and how do you deal with the perturbation inputs.


7. Could you please clarify why the baselines are different from the original SLICE paper?
     > 5.1 Claim 1: LIME suffers from inconsistencies due to sign flip and rank variance

8. I didn't find that you discussed the "Deletion and Insertion Game"? Could you please clarify?

**Strengths And Weaknesses:**

## Strengths
1. Overall, this manuscript is well-written.
2. This manuscript provides a method for generating perturbed images named GRID-LIME, and a metric GTO assessing the overlap between ground-truth and explanations.

## Weaknesses
1. Simply reproducing a paper doesn't automatically make this paper above the acceptance bar in TMLR, unless the manuscript delivers significantly meaningful arguments during the reproducing progress.
2. Some parts in the presentation needs further clarification. I'll specify in the requested changes.
3. The newly proposed GRID-LIME and GTO are not well justified.
5. This reproducibility work only evaluated resent50 and inceptionv3, this lack of the variety of architectures undermines the value of this work. I'll reserve until I see more architectures are included.

---

> ### Author Response · Authors · 2025-04-01
> **Response to Reviewer XQ5V (1/3)**
>
> We thank Reviewer XQ5V for their detailed review and constructive feedback on our reproducibility study of SLICE . We appreciate the acknowledgment of the paper's writing and the novelty introduced through GRID-LIME and the GTO metric. We are submitting this work as students for the Machine Learning Reproducibility Challenge (MLRC) 2025 and have worked to address the reviewer's valuable suggestions within the scope and resources available to us.
>
> We address each specific point below:
>
> 1. Weakness: Significance of Reproducibility Study
>
>    Reviewer Comment: "Simply reproducing a paper doesn't automatically make this paper above the acceptance bar in TMLR, unless the manuscript delivers significantly meaningful arguments during the reproducing progress."
>
>    Response: We agree that a reproducibility study should offer more than simple verification. We believe our study provides several significant contributions and insights derived directly from the reproduction process:
>
>    Critical Analysis: While confirming SLICE's consistency improvements over LIME, our work critically evaluates and challenges its claimed fidelity superiority. Our analysis using standard metrics (AOPC/AUC) and our proposed GTO metric reveals a more nuanced picture where SLICE's fidelity is debatable and potentially influenced by its perturbation strategy (Sections 5.4, 5.6, Appendix B).
>
>    Methodological Insights: We identified potential biases in fidelity evaluation stemming from different perturbation techniques (blur vs. masking). This led us to propose an alternative AOPC evaluation (Appendix B) for fairer comparison and the GTO metric (Section 4.4) as a perturbation-agnostic fidelity measure focused on segmentation alignment.
>
>    Practical Limitations: Our study highlights the significant computational cost of SLICE (Appendix D, Table 4), a crucial practical limitation not extensively detailed in the original work.
>
>    Addressing Gaps: We encountered incompleteness in the original authors' provided code (missing metric implementations), requiring us to re-implement key evaluation components (ASFE, ARS, CCM), which we have now made available (Section 6.2, 6.3).
>
>    Novel Proposals: The challenges and insights above directly motivated our proposals for GRID-LIME (addressing LIME's instability and SLICE's cost) and the GTO metric.
>
>    We have revised the Introduction (Section 1) and Discussion (Section 6) to better emphasize these findings and arguments derived from the reproduction effort.
>
> 2. Weakness: Presentation Clarity & Justification of GRID-LIME/GTO
>
>    Reviewer Comments: "Some parts in the presentation needs further clarification." / "The newly proposed GRID-LIME and GTO are not well justified." / "I searched for the motivations for proposing LIME-GRID and GTO but I couldn't see them. Could you please clarify? I would like to suggest the authors justify the motivations as earlier as possible in eg Introduction."
>
>    Response: We thank the reviewer for pointing out the need for clearer justification and motivation. We have significantly revised the Introduction (Section 1) to explicitly state the motivations early in the paper, linking them directly to the limitations observed during the SLICE reproduction:
>
>    GTO Motivation: Introduced because standard fidelity metrics like AOPC/AUC can be confounded by the choice of perturbation (blur in SLICE vs. masking in LIME). GTO offers a direct, potentially less biased evaluation based on alignment with ground-truth object boundaries.
>
>    GRID-LIME Motivation: Proposed to address two key issues identified: (1) LIME's explanation instability, partly stemming from the randomness of its quickshift segmentation, and (2) SLICE's significant computational overhead. GRID-LIME aims to improve stability over LIME by using a structured, model-informed grid while maintaining computational efficiency closer to LIME.
>
>    We believe these revisions make the rationale behind our contributions much clearer from the outset. Specific clarification points are addressed below.

---

> > ### Comment · Reviewer_XQ5V · 2025-04-14
> > **Thanks the efforts from authors.**
> >
> > I thank the authors for their efforts. My concerns are now mostly addressed.

---

> ### Author Response · Authors · 2025-04-01
> **Response to Reviewer XQ5V (2/3)**
>
> 3. Weakness: Lack of Architectural Variety
>
>    Reviewer Comment: "This reproducibility work only evaluated resent50 and inceptionv3, this lack of the variety of architectures undermines the value of this work. I'll reserve until I see more architectures are included." / "Section 4.4: Can you also include other architectures, eg, vision transformation?"
>
>    Response: We acknowledge that evaluating a wider range of architectures, such as Vision Transformers (ViTs), would indeed strengthen the generalizability of the findings. However, the primary goal of this work was to reproduce the experiments and validate the claims presented in the original SLICE paper. The original authors used ResNet50 and InceptionV3 as their evaluation models. Faithfully replicating their setup was our main focus. Extending the study to significantly different architectures like ViTs represents substantial additional work, requiring different preprocessing, potentially different hyperparameter tuning, and considerable computational resources, which was beyond the scope and resources available for this MLRC reproducibility study. We agree this is an important direction for future work but maintain that our current scope fulfills the requirements of a focused reproducibility study.
> 4. Requested Change: Justification of Perturbation Methods
>
>    Reviewer Comment: "I'm wondering the justification of various perturbation method. The perturbation sampling injects our priors (perhaps bias sometimes), does this prior/bias ensure the faithful explanation?"
>
>    Response: This is an excellent point regarding the foundations of perturbation-based explanation methods.
>
>    Justification: LIME, SLICE, and GRID-LIME all rely on perturbation (masking regions or blurring them) as a way to probe the black-box model's local sensitivity. By observing how the model's prediction changes when parts of the input are altered, these methods infer the importance of those parts. The underlying assumption is local linearity – that the complex model behaves somewhat linearly in the close vicinity of the input being explained.
>
>    Prior/Bias: The choice of perturbation does inject a prior or potential bias. Masking (setting pixels to zero/mean, as in LIME/GRID-LIME) assumes that removing a feature equates to its absence having a meaningful impact. Blurring (as in SLICE) assumes that reducing high-frequency information locally is a meaningful perturbation. Both methods generate inputs that might be out-of-distribution (OOD) for the original model.
>
>    Faithfulness: "Faithfulness" in this context refers to how well the local surrogate model (e.g., the linear model in LIME) approximates the black-box model's behavior in the local neighborhood defined by the perturbations. It doesn't guarantee global truth or perfect causal attribution. The bias introduced by the perturbation method can affect this local approximation. This is precisely why we found differing fidelity results depending on the metric (AOPC vs. GTO) and perturbation (masking vs. blur). Our alternative AOPC analysis (Appendix B) directly explores this by testing SLICE with LIME's perturbation method, and the GTO metric bypasses perturbation-based fidelity evaluation altogether. We have added discussion on this point in Section 6.
>
> 5. Requested Change: Section 4.1.3 Logic Clarification
>
>    Reviewer Comment: "Section 4.1.3: (A)Since LIME relies on color-space segmentation (quickshift), its explanations suffer from randomness. (B) GRID-LIME eliminates this variability by enforcing consistent superpixel regions... Could you please clarify the logic between (A) and (B)?"
>
>    Response: We apologize for the lack of clarity. The logic is as follows:
>
>    (A) LIME typically uses segmentation algorithms like quickshift. Quickshift groups pixels based on color and spatial proximity. Due to minor variations in implementation or initialization, or inherent sensitivity to small pixel value changes, running quickshift multiple times on the same image can produce slightly different superpixel boundaries. This inconsistency in the segmentation itself contributes to the run-to-run variability (randomness) of LIME's final explanations.
>
>    (B) GRID-LIME replaces this potentially random segmentation process with a deterministic, structured grid. Although the optimal grid size is chosen based on model sensitivity (Equation 3), once chosen, that grid structure is fixed for that image. All subsequent perturbation steps within the LIME framework use these exact same grid cells as superpixels. By removing the variability inherent in the quickshift segmentation step, GRID-LIME eliminates that specific source of randomness, thereby enforcing consistent superpixel regions across the explanation generation process for a given image. We have revised the wording in Section 4.1.3 to make this connection clearer.

---

> ### Author Response · Authors · 2025-04-01
> **Response to Reviewer XQ5V (3/3)**
>
> 6. Requested Change: Section 4.3 Hyperparameter Choices
>
>    Reviewer Comment: "Section 4.3: For SLICE, we train 1000 Ridge regression models... on 500 randomly generated perturbations... Can you clarify the choices of these settings?"
>
>    Response: These specific hyperparameter values (M=1000 Ridge models, N=500 perturbations per model) were adopted directly from the methodology described in the original SLICE paper. Our goal was to reproduce their results as faithfully as possible, hence, we used their specified settings. We have clarified this explicitly in the revised Section 4.3.
>
> 7. Requested Change: GRID-LIME Zeroing & Distribution Shift
>
>    Reviewer Comment: "In lime-grid, when you zero super pixels, you significantly change the distribution of input X... Please clarify how do you consider regarding this, and how do you deal with the perturbation inputs."
>
>    Response: The reviewer raises a crucial point about perturbation methods creating OOD inputs.
>
>    Consideration: This is an inherent characteristic and potential limitation of masking-based perturbation methods like standard LIME, which GRID-LIME adopts. The underlying assumption is that the model's response to this OOD input (where a feature is "removed") is indicative of that feature's importance for the original prediction. We acknowledge that this introduces a strong assumption, and the model's behavior on such inputs might not perfectly reflect its behavior on in-distribution data.
>
>    How Dealt With: GRID-LIME, like LIME, doesn't explicitly "deal with" the distribution shift beyond using these perturbed samples to fit the local surrogate model. SLICE attempts to mitigate this by using Gaussian blurring, creating arguably "smoother" and potentially less OOD perturbations compared to hard masking. Our study implicitly investigates the impact of this difference through:
>
>    - Comparing SLICE (blur) vs. LIME/GRID-LIME (masking) using standard metrics (Section 5.4).
>    - Our alternative AOPC analysis (Appendix B), where SLICE is evaluated using masking, allowing a direct comparison of the algorithms independent of the perturbation type.
>    - The GTO metric (Section 5.6), which assesses fidelity based on segmentation overlap, independent of the perturbation mechanism.
>
>    We have added a brief discussion of this OOD issue in Section 6.
>
> 8. Requested Change: Baselines Different from Original Paper
>
>    Reviewer Comment: "Could you please clarify why the baselines are different from the original SLICE paper?"
>
>    Response: Our primary baseline for comparison against SLICE is LIME, which is the same baseline used in the original SLICE paper. Our study directly compares SLICE to LIME across all metrics, mirroring the original paper's core comparison. GRID-LIME is not a baseline from the original paper; it is a novel method proposed within this reproducibility study as an alternative derived from our findings. We compare SLICE, LIME, and GRID-LIME against each other. We have ensured the LIME implementation follows standard procedures, and hyperparameters (Appendix C) are consistent for fair comparison between all three methods evaluated in our study.
>
> 9. Requested Change: Discussion of "Deletion and Insertion Game"
>
>    Reviewer Comment: "I didn't find that you discussed the 'Deletion and Insertion Game'? Could you please clarify?"
>
>    Response: The concepts commonly referred to as the "Deletion and Insertion Game" are indeed evaluated in our study, primarily through the AOPC and AUC metrics (Section 5.4, Figures 6 and 7, Appendix B).
>
>    Insertion: Measures how the model's prediction score increases as the most important positive superpixels are sequentially added back to a baseline (e.g., blurred or masked) image. A higher AOPC/AUC for insertion indicates better identification of relevant positive features.
>
>    Deletion: Measures how the model's prediction score decreases as the most important features (e.g., most negative superpixels using MoRF as in the original paper) are sequentially removed from the original image. A higher AOPC / lower AUC for deletion generally indicates better identification of influential features whose removal impacts the prediction.
>
>    Our AOPC/AUC evaluations follow the Most Relevant First (MoRF) strategy used in the original paper, which implements these insertion/deletion procedures. The captions for Figures 6 and 7 explicitly mention "insertion task" and "deletion task". We have added a sentence in Section 5.4 explicitly linking AOPC/AUC and MoRF to the evaluation of insertion and deletion performance.
>
> We hope these clarifications and revisions adequately address the reviewer's concerns. We appreciate the thorough feedback, which has helped improve the rigor and clarity of our study.

---

### Review · Reviewer_CUpd · 2025-03-19

**Summary Of Contributions:**

This work reproduces and evaluates a post-hoc model explainability method called SLICE. SLICE (Bora et al., 2024) enhances another method called LIME by incorporating adaptive Gaussian blurring and sign entropy-based feature elimination to improve explanation consistency. The authors of this work validate SLICE using images from datasets including MSCOCO, Oxford-IIIT Pets and PASCAL VOC (50 images from each dataset). They find SLICE can produce more consistent explanations. Moreover, by fidelity analysis, they find SLICE is influenced by perturbation techniques and its deletion strategy is worse than LIME on the proposed AOPC score. The authors additionally propose Ground Truth Overlap (GTO) metric to validate SLICE, and they find LIME often produces more segmentation-aligned attributions. Finally, they mention that SLICE is much slower than LIME, and they propose one additional method called GRID-LIME that replaces LIME’s segmentation-based superpixels with a structured grid, obtaining faster and more stable results.

**Audience:**

Yes

**Claims And Evidence:**

Yes

**Requested Changes:**

As mentioned in the weaknesses, the authors should replace all the figures in the paper with higher resolution images. Moreover, the authors should add additional experimental results to validate the reproduced code and also the value of their proposed GRID-LIME, since from current results, GRID-LIME has worse explanation consistency than SLICE and generates worse object boundaries.

**Strengths And Weaknesses:**

Strengths:
(1) The authors reproduce SLICE using pytorch, making the method easier to use in the future.

(2) The authors analyze SLICE in detail, including the segmentation-related performance and fidelity. They propose new metrics to evaluate explainability methods and find the directions for possible future improvements.

Weaknesses:
(1) The writing should be improved a lot. For example, all the figures have low resolutions and the details are difficult to read.

(2) The reproduced code is not validated. At least in one experiment the authors should provide the results done by the original SLICE code.

(3) The importance of GRID-LIME is not emphasized enough. For example, in the main paper the authors should have a table showing the inference speed of all the methods to validate the claim "GRID-LIME improves explanation stability while maintaining a computational cost closer to LIME than SLICE".

---

> ### Author Response · Authors · 2025-04-01
> **Response to Reviewer CUpd (1/2)**
>
> We sincerely thank the reviewer for their time and insightful feedback on our reproducibility study of SLICE. We appreciate the positive comments regarding our PyTorch implementation and the detailed analysis conducted. The constructive criticism has been invaluable in improving the quality and clarity of our manuscript. We are submitting this work as students for the Machine Learning Reproducibility Challenge (MLRC) 2025, and while we faced some resource limitations, we have strived to address the reviewer's concerns thoroughly in our revised version.
>
> Below, we address each point raised:
>
> 1. Weakness: Writing and Figure Resolution
>
>    Reviewer Comment: "The writing should be improved a lot. For example, all the figures have low resolutions and the details are difficult to read."
>
>    Response: We thank the reviewer for highlighting the issues with figure resolution and writing.
>
>    Figure Resolution: We acknowledge that the figures in the original submission were not of sufficient quality. In the revised manuscript, we have replaced all figures (Figures 1-14, plus figures in the appendices) with high-resolution versions. We believe this significantly improves readability and allows for better appreciation of the details in the plots and diagrams.
>
>    Writing: We have carefully proofread and revised the entire manuscript to improve clarity, conciseness, and overall flow. We have paid particular attention to ensuring that our claims, methods, and results are presented in a clear and understandable manner.
>
> 2. Weakness: Reproduced Code Validation
>
>    Reviewer Comment: "The reproduced code is not validated. At least in one experiment the authors should provide the results done by the original SLICE code."
>
>    Response: We agree with the reviewer that validating our reproduced code against the original implementation is crucial.
>
>    In the original submission (Section 6.1 "What was easy"), we mentioned validating our PyTorch implementation against the authors' provided pickle files for two specific images to confirm basic consistency. However, we understand the need for a broader comparison.
>
>    Running the original TensorFlow implementation for large-scale experiments was challenging due to its significantly longer runtime (~15 minutes per image on our hardware, as noted in Section 6.2 "What was difficult") and our resource constraints as students.
>
>    To address this concern as best as possible within these constraints, we have now added Appendix F: Results from original paper. This new appendix includes Figure 15, which shows the Kernel Density Estimation (KDE) plots for the Combined Consistency Metric (CCM) generated using the original authors' TensorFlow code for SLICE on the Oxford-IIIT Pets and Pascal VOC datasets (using both ResNet50 and InceptionV3).
>
>    This allows for a direct visual comparison between the results from the original code (Appendix F, Figure 15) and the results from our PyTorch implementation for SLICE (Main paper, Figure 5). The comparison shows highly similar distribution patterns, providing evidence that our PyTorch implementation faithfully reproduces the consistency behavior of the original SLICE method. We believe this addition substantially strengthens the validation of our codebase.

---

> > ### Author Response · Authors · 2025-04-01
> > **Response to Reviewer CUpd (2/2)**
> >
> > 3. Weakness: Emphasis and Validation of GRID-LIME
> >
> >    Reviewer Comment: "The importance of GRID-LIME is not emphasized enough. For example, in the main paper the authors should have a table showing the inference speed of all the methods to validate the claim 'GRID-LIME improves explanation stability while maintaining a computational cost closer to LIME than SLICE'. ... from current results, GRID-LIME has worse explanation consistency than SLICE and generates worse object boundaries."
> >
> >    Response: We appreciate the reviewer's feedback on the positioning and validation of GRID-LIME.
> >
> >    Computational Cost Validation: To provide concrete evidence for the computational efficiency claim, we have added Table 4 in Appendix D: Computational requirements and environmental impact. This table explicitly lists the average runtime (in minutes) and estimated carbon emissions for LIME, SLICE, and GRID-LIME across all tested datasets and models. The results clearly show that GRID-LIME's runtime is very close to LIME's and significantly faster (up to ~13x faster) than SLICE's runtime. We have added references to this appendix table in the main Discussion (Section 6) to quantitatively support the claim.
> >
> >    Emphasis and Value Proposition: The reviewer correctly observes that GRID-LIME does not outperform SLICE on consistency (Figure 5) and performs worse than LIME on the GTO metric (Figure 8). We acknowledge this and have revised the manuscript (particularly the Introduction in Section 1 and the Discussion in Section 6) to better frame GRID-LIME's contribution. We now position GRID-LIME less as a universally superior method, and more as a practical alternative that strikes a different trade-off. Specifically, GRID-LIME offers a significant improvement in consistency over standard LIME (as shown in Figure 5) while avoiding the substantial computational overhead associated with SLICE (validated in Appendix D, Table 4). This makes GRID-LIME a potentially valuable option in scenarios where computational resources are limited or SLICE's runtime is prohibitive. We present it as an outcome of our reproducibility study, demonstrating that addressing LIME's segmentation instability structurally (rather than via SLICE's iterative refinement) can improve consistency while maintaining efficiency. We believe this clearer framing accurately reflects its strengths and limitations based on our findings.
> >
> > Summary of Changes:
> >
> > - Replaced all figures with high-resolution versions.
> > - Revised text throughout for clarity and improved writing.
> > - Added Appendix F with CCM results generated using the original authors' SLICE (TensorFlow) code for validation purposes.
> > - Added Appendix D containing Table 4, which quantitatively compares the runtime and carbon emissions of LIME, SLICE, and GRID-LIME.
> > - Updated the Introduction (Section 1) and Discussion (Section 6) to better frame the trade-offs and value proposition of GRID-LIME, referencing the new computational cost data.
> >
> > We hope these revisions effectively address the reviewer's concerns and have strengthened the paper. We thank the reviewer once again for their valuable guidance.

---

> > > ### Comment · Reviewer_CUpd · 2025-04-18
> > > **Thank the authors for the revisions**
> > >
> > > Most of my concerns are addressed based on the revisions. Thank the authors for their efforts.

---

### Decision · Action_Editor_fBAu · 2025-05-03

**Recommendation:** Accept as is

**Comment:**

Some reviewers expressed concerns about the writing quality and the clarity of the figures. To expedite the publication process, I have decided to accept the paper as is, rather than requesting minor revisions. However, I will closely review the writing and figures upon final submission. The authors are strongly encouraged to ensure that the final version is clearly written and includes high-quality figures to facilitate prompt publication

**Audience:**

This paper would be interesting to a general ML audience, particularly those interested in explainability.

**Claims And Evidence:**

The paper works on reproducing the 2024 paper that presented SLICE as an improvement of LIME, one of the most cited explanability papers. The authors found the algorithm reproducible and proposed a new way of measuring interpretability. All the reviewers did a good job, pointed out weaknesses in the first review, and the authors addressed them in their rebuttal to the reviewers' satisfaction.